# Great Minds Think Alike: Contextual Tacit Communication for Decentralized LLM-Agent Cooperation

**Yue Pei** [1]  **Hongming Zhang** [2]  **Jiarui Guan** [3]  **Jusheng Zhang** [4]  **Liang Lin** [4][5]  **Haogang Zhu** [6][7]  **Ziliang Chen** [5]

## Abstract

Large language models (LLMs) are increasingly used as planners for cooperative embodied agents, but multi-agent settings amplify inconsistency under partial observability and make explicit communication costly or even unavailable. Many existing approaches rely on online message passing; when communication is removed, agents often fall back to independent local planning that suffers from miscoordination. We introduce Contextual Tacit Communication, a gradient-free protocol that aligns decentralized decisions with a joint LLM value score without explicit message actions. Our method measures context-conditioned value rectifications via residual banding to pinpoint miscoordination actions and amortizes the resulting coordination signals into a retrieval-augmented Tacit Rule Memory that provides prompt-level cooperation rules at execution time. Experiments on VIKI, C-WAH, and TDW-MAT show that our approach improves cooperation performance over baselines while reducing runtime overhead compared with communication-based methods.

## 1. Introduction

Large language model (LLM) agents are increasingly deployed in embodied environments to solve tasks that require *multi-agent cooperation*, such as household routines and object transport (Song et al., 2023; Wu et al., 2023; Mandi et al., 2024). However, embodied cooperation is fundamentally challenging under partial observability and decentralized execution: each agent only observes a local view and must act without access to teammates' intent or subgoals.

As a result, coordination failures are common. A dominant line of work addresses this by treating *communication* as part of the decision interface, allowing agents to exchange natural-language messages online to share observations and plans (Zhang et al., 2023; Liu et al., 2024).

Explicit communication can indeed improve coordination, but it introduces substantial runtime overhead: additional LLM calls for messaging and higher token costs, which can scale poorly as the number of agents grows. Moreover, explicit messaging may be infeasible in practice due to privacy constraints, or restricted interfaces where message actions are unavailable (Wang et al., 2020; Mao et al., 2020). These considerations motivate coordination mechanisms that *remove online message actions* while still recovering the benefits of information sharing.

A separate tradition in multi-agent learning studies coordination under the *centralized training, decentralized execution* (CTDE) paradigm (Oliehoek et al., 2008; 2016), typically by learning centralized critics and decentralized policies with differentiable objectives. While powerful, directly porting CTDE machinery to multi-LLM embodied agents is non-trivial: LLM agents are often large and closed-source, task outcomes are delayed, and end-to-end differentiable training or per-task fine-tuning can be impossible. This motivates a coordination protocol that updates no model parameters and can be applied to black-box LLM agents.

In this paper, we propose *Contextual Tacit Communication*: instead of exchanging explicit messages online, agents coordinate through an *implicit contextual signal* (agent-spec context) that conditions decentralized decision-making. Conceptually, tacit communication is realized when decentralized greedy choices under local observations and agent-spec context can align with a centralized team-level judgment (cf. Definition 2.1). The key question then becomes: *how do we construct such context so that decentralized agents can reliably coordinate without explicit messaging?*

Our answer is a contextual protocol that transfers centralized preferences into per-agent guidance. We use the LLM as a centralized critic and approximate the joint value by an additive representation of rectified per-agent values plus a shared baseline, then quantify mismatch using a *contextual*

---

[1]School of Artificial Intelligence, Beihang University [2]Institute of Automation, Chinese Academy of Sciences [3]Aalto University [4]Sun Yat-sen University [5]Peng Cheng Laboratory [6]School of Computer Science and Engineering, Beihang University [7]Zhongguancun Laboratory. Correspondence to: Ziliang Chen <c.ziliang@yahoo.com>.

*Proceedings of the 43rd International Conference on Machine Learning*, Seoul, South Korea. PMLR 306, 2026. Copyright 2026 by the author(s).

*residual*. We introduce *residual banding* and a projection-style fitting procedure that pinpoints which local actions contribute to joint-level inconsistency, yielding action-grounded bias signals. To make coordination efficient at test time, we further amortize these coordination signals into a Tacit Rule Memory (TRM). We distill the mined bias patterns into natural-language rules and store them with progress embeddings. During decentralized execution, each agent retrieves the most relevant rules by semantic similarity of its local progress and injects the retrieved rule texts as prompt context. This retrieval-triggered context provides an implicit coordination channel that requires no explicit message. We call our method *Tacit Alike Contextual Thinking* (*Tact*).

This design yields three intended benefits. First, implicit context serves as a communication substitute: it aligns decentralized greedy choices with centralized preference, thereby realizing *tacit communication* under partial observability. Second, residual banding provides precise, action-level supervision for mining miscoordination, and TRM enables amortized reuse of these patterns, reducing reliance on free-form dialogue and repeated coordination queries. Third, TRM rules are language-based, non-parametric, and triggered by semantic similarity of local progress, making them reusable across tasks and similar local progress states.

We validate our method on three benchmarks. On VIKI-bench (Kang et al., 2025), *Tact* improves over the decentralized baseline, indicating effective coordination. On embodied cooperation tasks C-WAH and TDW-MAT (Gan et al., 2022; Zhang et al., 2023), *Tact* reduces average steps and improves transport rates. *Tact* achieves strong task efficiency while substantially reducing inference overhead relative to explicit communication-based baseline (39.7% fewer in the 2-agent case) and exhibits more moderate cost growth as team size increases. In summary, our contributions are: (1) We formalize *contextual tacit communication* for multi-LLM agents under decentralized execution without explicit message actions and connect residual-banded consistency to near-optimal greedy selection. (2) We propose a gradient-free contextual protocol that uses residual banding to mine action-level miscoordination patterns and bias-guided distillation into *TRM* for efficient decentralized execution. (3) We demonstrate efficient cooperation on VIKI, C-WAH, and TDW-MAT environments.

## 2. Preliminary and Motivation

In this section, we describe the task notation and problem setup for better understanding.

**Multiple LLM-Agent Cooperation**. We study cooperative decision-making in embodied environments, where $N$ agents collaborate to accomplish a shared goal $g$ (e.g., 'prepare a meal'). At each step $t$, the agent $i \in$ $\{1, \ldots, N\}$ receives an agent-specific partial observation $o_{i,t}$ and maintains a history $h_{i,t} := (o_{i,1}, a_{i,1}, \ldots, o_{i,t})$. Each agent selects an action $a_{i,t}$ from a finite candidate set $\mathcal{A}_{i,t}$, and the team executes the joint plan $u_t = (a_{1,t}, \ldots, a_{N,t})$. The episode terminates at the horizon $T$ with a sparse outcome $y \in \{$ success, fail $\}$, and we write the team trajectory as $\tau := \left(g, \{h_{i,t}\}_{i=1}^{N}\right)_{t=1}^{T}$.

Recent multi-agent LLM systems treat *communication* as part of the decision interface by factoring the high-level action space as $\mathcal{A} = \mathcal{A}_{\text{env}} \times \mathcal{A}_{\text{comm}}$, where $\mathcal{A}_{\text{env}}$ contains environment actions and $\mathcal{A}_{\text{comm}}$ contains message actions that broadcast natural-language content to teammates. This explicit protocol can improve cooperation by sharing partial observations and intermediate plans, but it also adds runtime overhead (extra LLM calls and token cost) and may be infeasible under bandwidth, privacy, or interface constraints.

**Tacit Communication Protocol**. We study coordination without explicit online message actions: during execution, agents do not send natural-language messages to one another, and the communication component $\mathcal{A}_{\text{comm}}$ is disabled. Thus, each agent chooses only environment actions but may condition its decision on an agent-specific context $x_i$. The context $x_i$ serves as prompt-level guidance that can bias local decisions toward team-compatible behavior. The goal is to make independently selected local greedy actions agree with a centralized team-level preference.

**Definition 2.1** (Tacit Communication). Consider decentralized execution with explicit communication disabled, i.e., $\mathcal{A}_{i,t}^{\text{comm}} = \emptyset$, so that $a_{i,t} \in \mathcal{A}_{i,t}^{\text{env}}$ for all $i, t$. Let each agent $i$ select its action by locally maximizing a per-agent value $q_i(\tau_{i,t}, x_i, a)$ that conditions on its local progress $\tau_{i,t}$ and agent-spec context $x_i$, and let $Q_{\text{jt}}(\tau_t, u)$ be a centralized joint value over joint actions. We say the system realizes *tacit communication* if the decentralized greedy choices are consistent with the centralized optimum, i.e., for all $t$,

$$\arg\max_{u \in \mathcal{U}(\tau_t)} Q_{\text{jt}}(\tau_t, u) \cap \prod_{i=1}^{N} \arg\max_{a \in \mathcal{A}_{i,t}} q_i(\tau_{i,t}, x_i, a) \neq \emptyset. \tag{1}$$

Here $\prod_{i=1}^{N}$ denotes the Cartesian product of agents' locally greedy action sets.

Definition 2.1 is analogous to CTDE in that a team-level judgment is recovered through decentralized greedy choices. The difference is that CTDE typically learns centralized critics and decentralized policies with differentiable objectives, whereas our setting uses contextual signals for black-box LLM agents. Specifically, we replace online message actions with the contextual signals $\{x_i\}_{i=1}^{N}$ and inject these signals at inference time without updating model parameters, which is important for closed-source LLM agents.

## 3. Methodology

Definition 2.1 requires centralized judgment but decentralized execution. CTDE usually realizes this through end-to-end training with a centralized critic. In our setting, directly porting that machinery is non-trivial: we target a *contextual protocol* for LLM agents, without differentiable critics. Instead of learning a critic, we reuse the LLM as a black-box evaluator to provide a centralized scoring signal over candidate decisions. Inspired by prior work (Dong et al., 2025), given a trajectory progress $\tau_t$ and a candidate decision $u \in \mathcal{U}(\tau_t)$, we define a scalar value score by log-odds:

$$\mathcal{Q}_{\mathrm{LLM}}(\tau_t, u) = \log \frac{P(y_w \mid \tau_t, u)}{P(y_l \mid \tau_t, u)}, \qquad (2)$$

where $P(\cdot \mid \tau_t, u)$ denotes the LLM's conditional probability of emitting the designated outcome token, and $y_w := \texttt{success}$ and $y_l := \texttt{fail}$ are the designated success and failure outcome tokens, respectively. Larger $\mathcal{Q}_{\mathrm{LLM}}$ indicates stronger evidence that executing $u$ under progress $\tau_t$ will lead to eventual success. We instantiate this value function at both the team level and the individual level.

Once we can score joint plans centrally, the remaining question is how to translate these scores into per-agent action values. We then represent the joint value by additive per-agent values up to a shared baseline, which is motivated by the value-correction idea of prior work (Son et al., 2019; Rashid et al., 2020; Sunehag et al., 2017). We seek corrected values $Q_i'(\tau_{i,t}, x_i, \cdot)$ and a baseline $V(\tau_t)$ such that, for all $u \in \mathcal{U}(\tau_t)$,

$$\mathcal{Q}_{\mathrm{LLM}}(\tau_t, u) \approx \sum_{i=1}^{N} Q_i'(\tau_{i,t}, u_i) + V(\tau_t). \qquad (3)$$

$V(\tau_t)$ captures team-level information and coordination effects that are not attributable to any single agent, thereby compensating for the mismatch between joint and individual views and making greedy decentralized choices aligned with the joint ranking on $\mathcal{U}(\tau_t)$.

### 3.1. Measuring Tacit Communication

To implement tacit communication, the key methodological question is how to *transfer* centralized judgments into per-agent execution.

**Contextual biases and residual.** We first introduce *contextual biases* $b_i(\tau_{i,t}, a)$ to correct individual values:

$$Q_i'(\tau_{i,t}, a) := q_{\mathrm{LLM},i}(\tau_{i,t}, a) + b_i(\tau_{i,t}, a), \qquad (4)$$

where $q_{\mathrm{LLM},i}(\tau_{i,t}, a)$ is agent $i$'s LLM value score for action $a$ under its partial progress $\tau_{i,t}$. We introduce $b_i(\tau_{i,t}, a)$ as a correction that shifts this local score, forming a rectified utility $Q_i'(\tau_{i,t}, a)$, so that the resulting greedy choices

are compatible with the joint value function. To measure how close the rectified per-agent value come to reproducing the team-level value, we define the *contextual residual* as the difference between the additive surrogate $\sum_{i=1}^{N} Q_i'(\tau_{i,t}, u_i) + V(\tau_t)$ and the joint value function $\mathcal{Q}_{\mathrm{LLM}}(\tau_t, u)$:

$$\delta(\tau_t, u) := \sum_{i=1}^{N} Q_i'(\tau_{i,t}, u_i) + V(\tau_t) - \mathcal{Q}_{\mathrm{LLM}}(\tau_t, u). \qquad (5)$$

A small contextual residual means the rectified individual value can reproduce the centralized scoring under the current context, which is the key requirement for decentralized greedy choices to agree with the centralized judgment in Definition 2.1.

**Residual-band consistency over the scored candidates.** Because $\mathcal{Q}_{\mathrm{LLM}}$ and $q_{\mathrm{LLM},i}$ come from noisy black-box LLM queries, the additive surrogate cannot match the joint value exactly, so requiring $\delta(\tau_t, u) = 0$ for all $u$ is impractical, and we instead enforce a tolerance band. Let $u^\star := \arg\max_{u \in \mathcal{U}(\tau_t)} \mathcal{Q}_{\mathrm{LLM}}(\tau_t, u)$ be the best joint plan under the joint value function. We impose:

$$\delta(\tau_t, u^\star) = 0, \qquad (6)$$

$$|\delta(\tau_t, u)| \leq \varepsilon_{\mathrm{res}}, \qquad \forall u \in \mathcal{U}(\tau_t), \qquad (7)$$

where $\varepsilon_{\mathrm{res}} \geq 0$ sets the allowed mismatch: smaller $\varepsilon_{\mathrm{res}}$ forces a tighter match, while larger $\varepsilon_{\mathrm{res}}$ allows more deviation. Eq. (6) chooses the offset $V(\tau_t)$ so that the best joint plan $u^\star$ is matched exactly, and Eq. (7) requires every other scored candidate to be matched within $\pm\varepsilon_{\mathrm{res}}$.

**Projection-style bias fitting.** Because $\mathcal{Q}_{\mathrm{LLM}}(\tau_t, u)$ and $q_{\mathrm{LLM},i}$ come from black-box LLM queries, we fit $\{b_i\}$ using a simple projection-style procedure at each $\tau_t$. Given current $\{Q_i'\}$, set the baseline to satisfy the anchor constraint:

$$V(\tau_t) \leftarrow \mathcal{Q}_{\mathrm{LLM}}(\tau_t, u^\star) - \sum_{i=1}^{N} Q_i'(\tau_{i,t}, u_i^\star). \qquad (8)$$

Then sweep over $u \in \mathcal{U}(\tau_t)$ and compute $\delta(\tau_t, u)$. If $|\delta(\tau_t, u)| > \varepsilon_{\mathrm{res}}$, define the excess

$$e(\tau_t, u) := \delta(\tau_t, u) - \mathrm{clip}(\delta(\tau_t, u), -\varepsilon_{\mathrm{res}}, \varepsilon_{\mathrm{res}}),$$

which is distributed evenly across the actions:

$$b_i(\tau_{i,t}, u_i) \leftarrow b_i(\tau_{i,t}, u_i) - \frac{e(\tau_t, u)}{N},$$
$$Q_i'(\tau_{i,t}, u_i) \leftarrow q_{\mathrm{LLM},i}(\tau_{i,t}, u_i) + b_i(\tau_{i,t}, u_i). \qquad (9)$$

This update changes $\sum_i Q_i'(\tau_{i,t}, u_i)$ by exactly $-e(\tau_t, u)$, pushing $\delta(\tau_t, u)$ back to the nearest boundary of the band in Eq. (7). After each sweep, we recompute $V(\tau_t)$ using Eq. (8) to maintain the anchor $\delta(\tau_t, u^\star) = 0$. We stop when

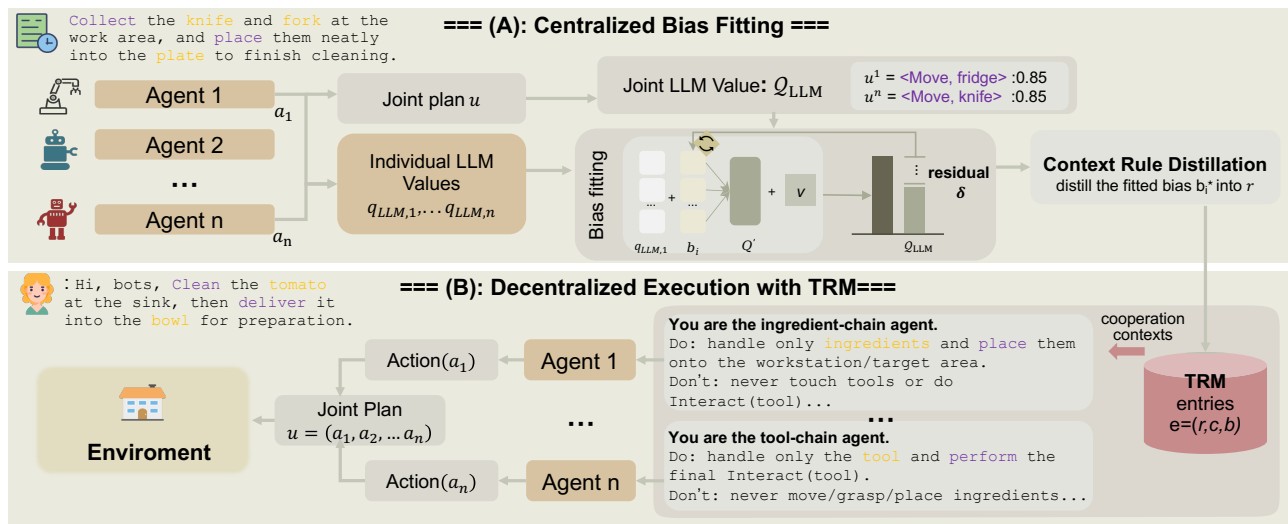

*Figure 1.* The overview of the framework. (A) A centralized joint LLM scorer evaluates candidate joint plans, while agents produce local values; we fit biases to align corrected local values with the joint judgment and distill them into TRM rules. (B) At execution time, each agent conditions on its local progress plus retrieved rule context from TRM to act without conflicts.

all candidates satisfy Eq. (7) or after $S_{\max}$ sweeps. The resulting bias $\{b_i\}$ is a context-conditioned coordination signal, which we distill into reusable prompt-level constraints in 4 to support decentralized execution without repeated joint scoring. Algorithm 1 is provided in Appendix B.

### 3.2. Theoretical discussion

In this section, we provide a theoretical guarantee for our residual-banding criterion: if the additive surrogate induced by the rectified per-agent values matches the joint LLM value within a bounded residual, then greedy action selection is near-optimal. This result gives a direct interpretation of the banding width $\varepsilon_{\mathrm{res}}$ as the worst-case loss in the joint LLM value.

Let $u^\star$ denote the best candidate under the joint evaluator, and let $\hat{u}$ be the greedy choice under the rectified additive representation, i.e., $\hat{u} := \arg\max_{u \in \mathcal{U}(\tau)} \sum_{i=1}^{N} Q_i'(\tau_i, u_i)$.

**Theorem 3.1** (Greedy guarantee from bounded residual). *If $\delta(\tau, u^\star) = 0$ and $\delta(\tau, u) \leq \varepsilon$ for all $u \in \mathcal{U}(\tau)$, then*

$$\mathcal{Q}_{\mathrm{LLM}}(\tau, u^\star) - \mathcal{Q}_{\mathrm{LLM}}(\tau, \hat{u}) \leq \varepsilon.$$

Theorem 3.1 formalizes the role of the residual: if the rectified additive values explain the joint LLM score up to an additive error $\varepsilon$ on the candidates, then the greedy decision induced by these values is guaranteed to incur at most $\varepsilon$ loss under the same joint value. This justifies using $\delta(\tau, u)$ as a contextual measure of tacit communication, which quantifies how well local context-conditioned preferences can reproduce the joint LLM judgment.

**Corollary 3.2** (Exact recovery as a special case). *Un-*

*der the conditions of Theorem 3.1 with $\varepsilon = 0$, $\hat{u} \in \arg\max_{u \in \mathcal{U}(\tau)} \mathcal{Q}_{\mathrm{LLM}}(\tau, u)$. In particular, if the maximizer is unique, then $\hat{u} = u^\star$.*

Corollary 3.2 is the $\varepsilon = 0$ special case of Theorem 3.1: driving the contextual residual to zero on the scored candidates makes the greedy plan induced by rectified local values recover the joint LLM-best candidate, which is the finite-candidate form of Definition 2.1.

**Connection to our implementation.** Our algorithm enforces a symmetric band $|\delta(\tau, u)| \leq \varepsilon_{\mathrm{res}}$ together with $\delta(\tau, u^\star) = 0$, which immediately implies the one-sided condition $\delta(\tau, u) \leq \varepsilon_{\mathrm{res}}$ required by Theorem 3.1. Hence $\varepsilon_{\mathrm{res}}$ admits a concrete interpretation: it upper-bounds the worst-case degradation in the joint LLM score caused by using the corrected decentralized values on the scored candidate set.

## 4. Tacit Rule Memory: Contextual Tacit Communication Guided by Retrieval Prompts

Tacit communication protocol is contextual since it coordinates agents implicitly by injecting cooperation-relevant context at inference time. To realize such contextual, implicit cooperation, we distill the fitted per-agent bias $b_i^\star(\tau_{i,t}, \cdot)$ into reusable context that LLM agents can condition on during decentralized execution. $b_i^\star$ provides an explicit coordination signal: it indicates which actions should be encouraged or discouraged under the current local progress. We amortize this signal into a retrieval-augmented Tacit Rule Memory (TRM) of natural-language texts that

agents can condition on at execution time with communication disabled. By expressing coordination as reusable, high-level rules and retrieving them by semantic similarity of local progress, the protocol can reuse coordination patterns when a new local state is semantically similar to stored examples, without task-specific parameter updates. We provide the overview of our framework in Figure 1.

**Building TRM entries**. For each agent $i$ and a local progress snapshot $\tau_i$ encountered during bias fitting, we form an entry $e = (r, \mathbf{c}, \mathbf{b})$: (i) $\mathbf{c}$ is an embedding of the local progress: we form a retrieval key text from the task description, the robot id, and an observation snippet, and encode it by mean-pooling the last-layer hidden states of a local text encoder; (ii) $\mathbf{b}$ is a *bias dict* that stores the fitted per-action corrections $b_i^\star(\tau_i, \cdot)$ obtained from the residual-banding procedure in Section 3.1, kept as a sparse dictionary over the selected actions; and (iii) $r$ is a natural-language rule text. Here $b_i^\star(\tau_i, a)$ is a directional correction that aligns the local value with the joint value: a positive bias means the local evaluator *underestimates* action $a$'s contribution to team success, so $a$ should be encouraged; a negative bias means the opposite, so $a$ should be avoided. Accordingly, we select a small set of actions with the most positive/negative bias values in $b_i^\star(\tau_i, \cdot)$, summarize them into an `encourage`/`avoid` supervision list, and prompt a rule-generation LLM to produce a rule text $r$ that explains *what to prioritize and what to avoid* under this type of progress. We store $r$ together with $\mathbf{c}$ and the sparse dictionary $\mathbf{b}[a] \approx b_i^\star(\tau_i, a)$ for the selected actions.

**Online retrieval and prompting.** TRM is built during an offline warm-up phase: the bias fitting and rule distillation described above are run on trajectories, and the resulting rule entries are written into memory. At execution time, TRM is static: no memory writing is performed and agents only read from it. Retrieval is by semantic similarity of local progress, so when two progress states have similar key-text embeddings the rule text can be reused across episodes and tasks. At the execution step $t$, the agent $i$ computes a query embedding from its current local progress $\tau_{i,t}$ and retrieves the most similar entries by embedding similarity. Let $\hat{x}_{i,t}$ denote the context of the retrieved rule texts. The agent then acts using its standard prompt augmented with $\hat{x}_{i,t}$. This realizes the agent-spec context in Definition 2.1. We provide an illustrative example in Appendix C.1.

**Diagnosing transfer of TRM rules.**

We analyze whether retrieved rules can transfer the distilled directional guidance to new local progress states. We do so by checking whether the rules recover the *direction* of the fitted bias on an action.

**Proposition 4.1** (TRM direction-mismatch decomposition)**.**
*Fix an action $a$ at some step and let $s^\star(a) \in \{-1, +1\}$ be the target direction given by the fitted bias ($+1$ encourages $a$,*

$-1$ *discourages $a$). TRM retrieves several memory entries. If the retrieved preferences for $a$ exist and all agree in sign, denote this common sign by $\tilde{s}(a) \in \{-1, +1\}$; otherwise set $\tilde{s}(a) = 0$. Let $\hat{s}(a) \in \{-1, +1\}$ be the direction actually induced after prompting with the retrieved rules. Define the events*

$$E_{\text{gap}} := \{\tilde{s}(a) = 0\},$$
$$E_{\text{dir}} := \{\tilde{s}(a) \neq 0, \tilde{s}(a) \neq s^\star(a)\},$$
$$E_{\text{comp}} := \{\tilde{s}(a) = s^\star(a), \hat{s}(a) \neq \tilde{s}(a)\}. Then$$

$$\Pr[\hat{s}(a) \neq s^\star(a)] \leq \Pr[E_{\text{gap}}] + \Pr[E_{\text{dir}}] + \Pr[E_{\text{comp}}].$$

$E_{\text{gap}}$ means TRM yields *no consistent* directional signal for $a$. $E_{\text{dir}}$ means TRM yields a consistent direction but it is *wrong* (disagrees with $s^\star(a)$); $E_{\text{comp}}$ means TRM yields the *correct* direction but the prompted agent does not comply. The inequality is a union-bound decomposition: a direction mistake $\hat{s}(a) \neq s^\star(a)$ can only happen via (i) no consistent retrieved guidance ($E_{\text{gap}}$), (ii) consistent-but-wrong guidance ($E_{\text{dir}}$), or (iii) non-compliance to a correct unambiguous rule ($E_{\text{comp}}$).

The dominant failure is typically $E_{\text{gap}}$: retrieval may not cover the relevant action or may return conflicting rules, producing no consistent direction. Therefore, the decomposition identifies $E_{\text{gap}}$ as a key bottleneck for transfer. TRM mitigates this gap by retrieving bias-grounded rules from semantically similar local progresses. Bias-grounded rule construction further reduces conflicts and stabilizes the direction across retrieved entries. The remaining terms ($E_{\text{dir}}$ and $E_{\text{comp}}$) correspond to wrong-sign retrieval and in-context non-compliance, which are largely model/interface dependent. The formal proof is given in Appendix A.

## 5. Experiments

### 5.1. Experimental Setups

**Benchmarks**. We evaluate our method in decentralized embodied environments that require multi-agent coordination under partial observations. **VIKI-Bench** is a hierarchical benchmark for embodied multi-agent collaboration with structured supervision for planning and cooperation. We use its VIKI-L2 planning tasks, which directly probe division of labor and action ordering (Kang et al., 2025). Since VIKI-L2 provides global visual observations while our focus is decentralized coordination, we modify VIKI-L2 to restrict agents to local observations for decentralized planning; full details are deferred to Appendix D.2.5. **Communicative Watch-And-Help (C-WAH)** contains episodes spanning five household activity types, where agents are jointly tasked with completing long-horizon household routines. Each task is specified by 3-5 predicates with a 250-step horizon and is evaluated under both symbolic and egocentric visual observations (Zhang et al., 2023). **ThreeDWorld Multi-Agent**

*Table 1.* Main results on VIKI. We report $\text{ACC}_{\text{ID}}$, $\text{ACC}_{\text{OOD}}$ (higher is better). Qwen2.5-VL-7B-SFT is obtained by answer fine-tuning on the training split: we supervise the model to directly output the target action plan given the same inputs.

| Metric | GPT-4o | | | | Qwen2.5-VL-72B-Instruct | | | | Qwen2.5-VL-7B-Instruct-SFT | | | |
|---|---|---|---|---|---|---|---|---|---|---|---|---|
| | P-cen | P-dec | Tact-b | Tact | P-cen | P-dec | Tact-b | Tact | P-cen | P-dec | Tact-b | Tact |
| $\text{ACC}_{\text{ID}} \uparrow$ | **20.5** | 2.4 | - | 19.7 | 9.0 | 0.0 | 5.4 | **16.4** | **96.5** | 5.2 | 77.4 | 89.2 |
| $\text{ACC}_{\text{OOD}} \uparrow$ | 9.6 | 0.0 | - | **10.5** | 1.2 | 0.0 | 0.0 | **5.4** | 23.6 | 2.0 | 3.6 | **30.2** |

*Table 2.* Average steps ($\downarrow$) on the C-WAH benchmark under visual (Vis. Obs.) and symbolic (Sym. Obs.) perception. Lower is better. MHP* uses a single agent, while all others adopt two agents.

| | MHP | | Qwen2.5-72B-Instruct | | | | GPT-4o | | | |
|---|---|---|---|---|---|---|---|---|---|---|
| | MHP* | MHP | CoELA | ProAgent | RoCo | *Tact* | CoELA | ProAgent | RoCo | *Tact* |
| Vis. Obs. | 141.0 | 103.0 | 96.1 | 105.3 | 110.4 | **94.7** | 90.8 | 98.2 | 112.5 | 92.4 |
| Sym. Obs. | 111.0 | 75.0 | 60.2 | 65.0 | 70.7 | **53.1** | 50.7 | 65.3 | 63.8 | **46.8** |

**Transport (TDW-MAT)** is built on the TDW simulation platform and requires agents to search for targets and transport objects to designated destinations, either by hand or using containers that increase carrying capacity. It contains 24 episodes across food and household-item transport with a 3000-frame budget (Zhang et al., 2023). More environment descriptions are detailed in Appendix D.1.

**Evaluation Metrics**. On VIKI-L2, we follow the official evaluation protocol and report identification accuracy on in-distribution and out-of-distribution splits, denoted as $\text{ACC}_{\text{ID}}$ and $\text{ACC}_{\text{OOD}}$. For Level-2 task planning, a predicted plan is counted as correct if it is feasible under the environment dynamics and no longer than the ground-truth plan. On C-WAH, we report Average Steps (lower is better), the number of environment steps required to satisfy all task predicates. On TDW-MAT, we report Transport Rate (TR) (higher is better), defined as the fraction of required objects delivered within 3000 frames, reported for food, and stuff.

**Implementation**. We instantiate agents with both closed-source and open-source LLM backbones: GPT-4o via the OpenAI API (Hurst et al., 2024), and Qwen2.5-VL-72B-Instruct / Qwen2.5-72B-Instruct (Bai et al., 2023). We use a maximum of 256 output tokens for each agent generation step. For the offline warm-up stage, the LLM critic and the log-odds value $\mathcal{Q}_{\text{LLM}}$ are computed with a local open-source backbone. The joint candidate set $\mathcal{U}(\tau_t)$ is constructed as a small structured subset rather than the full Cartesian product to keep centralized scoring tractable; details of the critic computation and candidate construction are provided in Appendix D.2.2 and Appendix D.2.3. The TRM construction and the online retrieval pipeline follow the procedure described in Appendix D.2.4.

**Baselines**. For VIKI-L2, the official planner adopts a global observation and outputs action sequences for all agents; we refer to this setting as a *centralized planner* (P-cen). To enable a fair comparison under partial observability, we

adapt this planner to operate with partial observations and per-agent planning, which we call a *decentralized planner* (P-dec). We also include a supervised fine-tuned variant trained under the global-view setting (P-cen-sft). These planner variants serve as reference baselines on VIKI-L2. For C-WAH and TDW-MAT, we compare against both classical hierarchical planners and LLM-based cooperative agents. The classical baselines include a MCTS-based hierarchical planner (MHP) (Puig et al., 2020) and a heuristic rule-based hierarchical planner (RHP) (Gan et al., 2022). We additionally evaluate representative LLM multi-agent systems: CoELA (Zhang et al., 2023), which coordinates via explicit natural-language communication; RoCo (Mandi et al., 2024), which performs centralized LLM planning with multi-turn team deliberation; and ProAgent (Zhang et al., 2024), a decentralized LLM planner that reasons about teammate intentions.

### 5.2. Main Results

Table 1 summarizes VIKI-L2 planning performance. A strong centralized planner based on supervised fine-tuning (P-cen with Qwen2.5-VL-7B-Instruct-SFT) achieves the highest in-distribution accuracy ($\text{ACC}_{\text{ID}}$=96.5). In contrast, restricting planning to partial observations causes a sharp degradation for the decentralized baseline (P-dec), on both ID and OOD splits. In VIKI-L2, much of the partial observability arises from unobserved teammate intent and intermediate subgoals, so an agent cannot reliably anticipate complementary actions required for coordinated subtask completion. Across all three backbones, *Tact* consistently outperforms P-dec, indicating implicit coordination from context conditioning. *Tact-b* is a variant of *Tact* that applies only numeric rectification using the result from the bias table without rule constraints; it can improve $\text{ACC}_{\text{ID}}$ in some cases but yields weaker and less stable gains than the full *Tact*. Tables 2 and 3 report embodied cooperation on C-WAH and TDW-MAT. Overall, *Tact* achieves the best

*Table 3.* Transport rate (%, ↑) on TDW-MAT with oracle perception. Higher is better. RHP* uses a single agent, while all others adopt two agents.

| | RHP | | Qwen2.5-72B-Instruct | | | | GPT-4o | | | |
|---|---|---|---|---|---|---|---|---|---|---|
| | RHP* | RHP | CoELA | ProAgent | RoCo | *Tact* | CoELA | ProAgent | RoCo | *Tact* |
| Food (↑) | 52.0 | 76.0 | **84.5** | 80.6 | 82.0 | 82.8 | 85.4 | 78.5 | **85.5** | 85.2 |
| Stuff (↑) | 49.0 | 74.0 | 76.3 | 78.7 | 79.2 | **80.3** | 84.0 | 83.6 | 85.1 | **87.8** |
| Avg. (↑) | 50.5 | 75.0 | 80.4 | 79.7 | 80.6 | **81.6** | 84.7 | 81.1 | 85.3 | **86.5** |

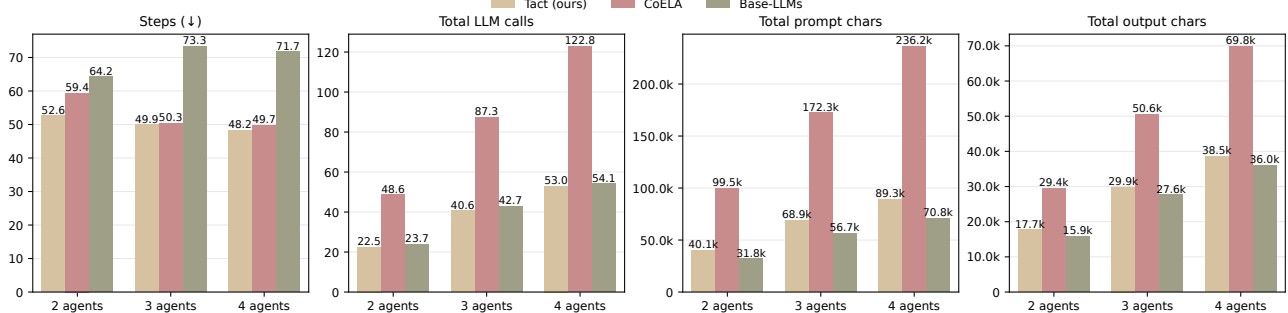

*Figure 2.* Online inference cost and scalability on C-WAH as the team size increases (up to 4 agents). We compare *Tact* with a communication-based baseline (CoELA) and an uncoordinated multi-agent baseline (Base-LLMs). We report episode-level totals aggregated across agents: number of LLM calls, prompt characters, and generated output characters, together with task efficiency (steps).

or near-best performance across settings. On C-WAH, *Tact* achieves the best performance under symbolic observations, and remains close to the best communication-based baseline under visual observations. On TDW-MAT, *Tact* achieves the highest transport rate on stuff transport and remains competitive on food transport.

**Cost Analysis and Agent Scaling**. To quantify the online inference overhead induced by different collaboration mechanisms, we compare three baselines as shown in Figure 2: *Tact* (ours); CoELA, where agents exchange messages by communication; and Base-LLMs, where multiple agents independently invoke the LLM backbone without communication. All baselines run on the same machine with the identical LLM backbone on C-WAH environment. We measure *total LLM calls*, *total prompt chars*, and *total output chars* (generated text), aggregated at the episode level across all agents; we evaluate team sizes up to 4 agents. Our method achieves the best task efficiency while also minimizing inference cost, which reduces LLM usage substantially compared to CoELA (e.g. 17.7k vs. 29.4k output chars, decreased by 39.7% in the 2-agent case), indicating lower overhead during online decentralized execution while retaining effective cooperation. As the number of agents increases to 3/4, base LLM without coordination degrades as agents increase. CoELA further reduces task steps but incurs a multiplicative increase in LLM overhead, whereas our method improves task efficiency with markedly more moderate growth in inference cost.

**Results Discussions**. Overall, we summarize the results

with three observations: **(1)** *Tact* can realize *tacit communication*. Purely local planning is fragile under partial observability (e.g., missing teammate intent on VIKI-L2), while injecting *implicit contextual signals* can align decentralized greedy choices with the centralized preference. **(2)** Our method attains stronger cooperation efficiency with lower runtime overhead compared with baselines. The advantage comes from precise mining of miscoordination patterns via residual banding and amortized reuse via TRM, avoiding per-step free-form dialogue generation and reducing the need for repeated coordination queries. **(3)** *Tact* provides reusable language-based coordination via contextual rule memory. By distilling numeric rectification into retrieval-triggered rules that activate by semantic similarity of local progress, the resulting guidance is non-parametric. This makes the coordination knowledge naturally reusable across task variants and states, complementing CTDE-style coordination that relies on differentiable training and can be less portable in black-box LLM settings.

### 5.3. Ablation

**Module Contributions.** We conduct ablations on VIKI-L2 to isolate the contribution of each module. Table 4 compares five variants: **A1** removes TRM rule contexts at test time and selects actions using the numeric rectified value function. **A2** removes residual banding and replaces it with coarse heuristic supervision for rule synthesis; **A3** keeps retrieval but drops bias supervision, making rule generation free-form; **A4** is the base LLM planner; and **A5** is

*Table 4.* Ablation on module contributions. Marks indicate the component is enabled. We report $\text{ACC}_{\text{ID}}$ and $\text{ACC}_{\text{OOD}}$.

| Exp | Base Planner | Residual Band. | Bias Sup. | TRM Context | $\text{ACC}_{\text{ID}}$ ↑ | $\text{ACC}_{\text{OOD}}$ ↑ |
|-----|--------------|----------------|-----------|-------------|------------|-------------|
| A1 | ✓ | ✓ | ✓ | | 77.4 | 3.6 |
| A2 | ✓ | | ✓ | ✓ | 10.4 | 2.5 |
| A3 | ✓ | ✓ | | ✓ | 8.5 | 1.2 |
| A4 | ✓ | | | | 5.2 | 2.0 |
| A5 | ✓ | ✓ | ✓ | ✓ | 89.2 | 30.2 |

the full method. Overall, the full method (**A5**) performs best. Removing TRM rule context at test time (**A1**) substantially degrades performance (e.g., $\text{ACC}_{\text{ID}}$ drops from 89.2 to 77.4), highlighting TRM as the mechanism that amortizes the mined coordination signals into reusable rule context for decentralized execution. Moreover, both **A2** and **A3** collapse relative to **A5**, showing that simply appending rule text is insufficient. Without residual banding (**A2**), the supervision driving rule synthesis becomes too coarse to resolve coordination ambiguity, yielding weak and unreliable constraints. Without bias supervision (**A3**), rule generation loses an action-grounded signal, making the resulting guidance less specific. These ablations indicate that robust gains require *both* residual-banded bias fitting and TRM-based cooperation context to support decentralized execution.

**Sample Efficiency Comparison.** Since *Tact* uses an offline trajectory-dependent stage to fit biases and construct TRM, we further compare it with training-based RL baselines under matched or comparable trajectory budgets on VIKI-L2. We train a decentralized RL baseline, denoted as Dec-RL-ZERO, where each agent predicts actions solely from its own local observation. Dec-RL-ZERO is trained on Qwen2.5-VL-3B using GRPO (Shao et al., 2024) for 15 epochs with batch size 256 and rollout size $n = 5$, using the same task trajectories as *Tact*.

*Tact* and Dec-RL-ZERO use the same base model without SFT to ensure a controlled comparison under the same model capacity and data budget. VIKI-R-ZERO is the centralized RL method without the SFT stage; we include the official VIKI-R-ZERO 3B/7B results as centralized RL references. To make the accounting explicit, for RL methods, sampled LLM outputs are counted as trajectories × epochs × GRPO rollout size × number of agents. For *Tact*, we count offline value-query outputs, including $\sum_t |\mathcal{U}(\tau_t)|$ joint-action scoring queries and per-agent local value queries. As shown in Table 5, *Tact* uses roughly one quarter of the sampled outputs required by Dec-RL-ZERO while achieving higher in-distribution accuracy under the same trajectory budget. This suggests that residual-based bias fitting and TRM distillation can extract coordination signals more sample-efficiently than direct decentralized RL training in this zero-SFT setting.

**Sensitivity to $\varepsilon_{\text{res}}$.** We study the effect of $\varepsilon_{\text{res}}$, which controls a *residual band*. We quantify concordance using value regret: the joint-value gap between the centralized reference $u^{\text{cen}} = \arg\max_{u \in \mathcal{U}(\tau)} \mathcal{Q}_{\text{LLM}}(\tau, u)$ and the decentralized choice $u^{\text{dec}} = \arg\max_{u \in \mathcal{U}(\tau)} \sum_i Q_i'(\tau_i, u_i)$: $\text{ValueRegret}(\tau) = \mathcal{Q}_{\text{LLM}}(\tau, u^{\text{cen}}) - \mathcal{Q}_{\text{LLM}}(\tau, u^{\text{dec}})$. As $\varepsilon_{\text{res}}$ increases, value regret also increases, indicating that a looser residual band yields weaker concordance: the decentralized choice deviates further from the centralized greedy selection under the same candidate set. We also report the number of sweeps to satisfy the residual band. Consistent with the intended role of $\varepsilon_{\text{res}}$, the required sweeps decrease monotonically as the band is relaxed, reflecting reduced fitting effort under a looser tolerance. Finally, we report the task accuracy $\text{ACC}_{\text{ID}}$. $\varepsilon_{\text{res}} = 0.5$ achieves the best accuracy, while overly loose ($\varepsilon_{\text{res}} \geq 1.0$) degrades performance. Intuitively, too small $\varepsilon_{\text{res}}$ increases correction effort and can over-constrain decentralized behavior, whereas too large $\varepsilon_{\text{res}}$ allows larger coordination errors.

## 6. Related Work

**LLM-based Agents**. Recent work studies LLMs as autonomous agents that interact with tools and environments (Yang et al., 2023; Liu et al., 2023). LLMs can serve as high-level planners and reasoners for embodied decision-making (Wu et al., 2023), often via prompt engineering, reflection, and tool use to bridge perception and action. Beyond using LLMs as direct planners, an emerging direction treats LLMs as critics to provide long-horizon guidance. For example, LAC (Dong et al., 2025) computes action values from token-level logits of designated outcomes and improves policies in an actor-critic style. Related work such as ReAd (Zhang et al., 2025) further introduces learned advantage feedback to refine multi-agent plans with fewer LLM queries. Our method also uses LLM logits to define value scores, but focuses on a different problem: transferring centralized joint preferences into decentralized coordination signals. Our log-odds value can be viewed as an outcome preference over candidate decisions (Bai et al., 2026).

**Value Decomposition and CTDE**. A large body of cooperative MARL realizes coordination under centralized training with decentralized execution by factorizing a joint value into per-agent utilities. VDN (Sunehag et al., 2017) uses an additive factorization, QMIX (Rashid et al., 2020) relaxes it to a

*Table 5.* Sample efficiency comparison on VIKI-L2 under the zero-SFT setting. *Tact* and Dec-RL-ZERO use the same base model.

| Method | Fitting Traj. | Total Env. Steps | Sampled LLM Outputs | $ACC_{ID}$ ↑ | $ACC_{OOD}$ ↑ |
|---|---|---|---|---|---|
| *Tact* (3B) | 3,765 | 26,004 | 158,600 | 8.45 | 0.00 |
| Dec-RL-ZERO (3B) | 3,765 | 26,004 | 578,550 | 0.72 | 0.00 |
| VIKI-R-ZERO (3B) | 7,196 | 48,523 | 539,700 | 0.00 | 0.00 |
| VIKI-R-ZERO (7B) | 7,196 | 48,523 | 539,700 | 0.17 | 0.00 |

monotonic mixing network, and QTRAN (Son et al., 2019) further enlarges the representable class of joint values. Our additive surrogate in Eq. (3) borrows this decomposition view, but differs in two essential ways: (i) it is *gradient-free*, operating over a finite candidate set rather than learning a differentiable mixing network; and (ii) it uses black-box LLM scores obtained by querying, rather than a trained critic optimized end-to-end. Thus, the decomposition is used as an inference-time consistency criterion for coordination, not as a parameterized value-learning architecture.

**Memory and Experience Distillation for LLM Agents**. Another relevant line equips LLM agents with memory or distilled experience that can be reused at inference time. Reflexion (Shinn et al., 2023) stores verbal self-reflections, and Generative Agents (Park et al., 2023) maintain a retrievable memory stream; ExpeL (Zhao et al., 2024) and AutoGuide (Fu et al., 2024) distill past trajectories into natural-language insights or state-aware guidelines, while SEER (Cui et al., 2025) recalls stepwise experience from a continually updated pool. Our Tacit Rule Memory is related to this family of retrieval-based experience reuse, but differs in how its rules are grounded. Rather than storing free-form reflections, TRM distills residual-banded, action-level bias signals into rule texts, so each retrieved rule carries directional and decision-relevant guidance.

**Multi-Agent Cooperation**. One line of LLM-based multi-agent systems coordinates agents through explicit online communication. This connects to a longer tradition of learned communication in MARL, where agents learn what messages to send through differentiable or discrete channels (Foerster et al., 2016; Sukhbaatar et al., 2016). CoELA (Zhang et al., 2023) builds cooperative embodied agents with a dedicated communication module to exchange messages during execution, while CaPo (Liu et al., 2024) improves efficiency via meta-plan generation and progress-adaptive multi-turn discussions. RoCo (Mandi et al., 2024) similarly relies on dialectic multi-turn dialogue among robots to jointly reason about strategies and refine plans. These approaches can be effective but inherently require message actions and incur additional LLM calls that scale with team size. Our work replaces online messaging with implicit contextual coordination by injecting retrieved rules, enabling cooperation without per-step dialogue.

Another line of work attempts to cooperate without direct

*Table 6.* Sensitivity to the $\varepsilon_{res}$ on VIKI-L2. We report (i) ValueRegret, the joint-score gap between the centralized greedy choice and the decentralized choice (lower is better), (ii) the number of projection sweeps required to satisfy the residual constraint (lower is better), and (iii) task accuracy $ACC_{ID}$.

| $\varepsilon_{res}$ | Value regret | Num of sweeps | $ACC_{ID}$ |
|---|---|---|---|
| 0.2 | 0.55 | 6.54 | 88.4 |
| 0.5 | 0.62 | 4.84 | 89.2 |
| 1.0 | 0.74 | 3.93 | 87.8 |
| 2.0 | 0.92 | 2.90 | 82.6 |

messaging by modeling teammates or adapting to interaction. ProAgent (Zhang et al., 2024) predicts teammates' intentions and coordinates via belief correction, and related work explicitly infers belief states for collaboration (Li et al., 2023), while LIET (Li et al., 2025) adapts multi-agent LLM planning through individual utility learning and team-level evolution. From an information-design view, an informed agent can also shape teammates' decisions through committed signaling rather than explicit dialogue (Bai et al., 2024). Our method does not require explicit teammate modeling or parameter updates. Instead, residual banding mines action-level miscoordination patterns, and TRM amortizes them into reusable contextual rules for decentralized execution.

## 7. Conclusion and Limitations

We presented *Contextual Tacit Communication*, a gradient-free coordination protocol that enables decentralized LLM agents to cooperate without explicit online messaging. By using residual banding to identify miscoordination patterns and amortizing them into a retrieval-augmented Tacit Rule Memory, our approach provides reusable contextual guidance for efficient decentralized execution and cooperation.

Our approach removes explicit message actions by relying on implicit contextual guidance, but this design is not universally optimal. In tasks where success critically depends on exchanging fresh, high-bandwidth, or rapidly changing private observations (e.g., tight real-time negotiation), communication-based methods can be stronger because they can directly transmit intent and newly discovered information during execution. In addition, our Tacit Rule Memory is obtained via an offline rule-mining stage that incurs implicit costs; therefore, our method is best viewed as trading an offline investment for reduced execution overhead.

## Acknowledgements

We thank the anonymous reviewers for their constructive comments and suggestions, which helped improve the clarity and presentation of this work. We are also grateful to our colleagues for helpful discussions and to all co-authors for their dedicated efforts throughout the project. This work was supported by New Generation Artificial Intelligence-National Science and Technology Major Project 2025ZD0123401.

## Impact Statement

This paper presents work whose goal is to advance the field of Machine Learning. There are many potential societal consequences of our work, none which we feel must be specifically highlighted here.

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

# A. Proofs of Theoretical Results

## A.1. Proof of Theorem 3.1 and Corollary 3.2

*Proof of Theorem 3.1.* The proof follows the value-decomposition argument of (Son et al., 2019). Fix a trajectory progress $\tau$ and candidate set $\mathcal{U}(\tau)$. Recall the residual definition:

$$\delta(\tau, u) := \sum_{i=1}^{N} Q_i'(\tau_i, u_i) + V(\tau) - \mathcal{Q}_{\text{LLM}}(\tau, u), \qquad \forall u \in \mathcal{U}(\tau).$$

Rearranging gives, for any $u$,

$$\mathcal{Q}_{\text{LLM}}(\tau, u) = \sum_{i=1}^{N} Q_i'(\tau_i, u_i) + V(\tau) - \delta(\tau, u). \tag{10}$$

Let $u^\star$ be a maximizer of $\mathcal{Q}_{\text{LLM}}(\tau, \cdot)$ and let

$$\hat{u} \in \arg \max_{u \in \mathcal{U}(\tau)} \sum_{i=1}^{N} Q_i'(\tau_i, u_i).$$

Since $V(\tau)$ does not depend on $u$, $\hat{u}$ also maximizes $\sum_i Q_i'(\tau_i, u_i) + V(\tau)$ over $\mathcal{U}(\tau)$, hence

$$\sum_{i=1}^{N} Q_i'(\tau_i, \hat{u}_i) \geq \sum_{i=1}^{N} Q_i'(\tau_i, u_i^\star). \tag{11}$$

Using (10) and the assumption $\delta(\tau, u^\star) = 0$, we compute:

$$\mathcal{Q}_{\text{LLM}}(\tau, u^\star) - \mathcal{Q}_{\text{LLM}}(\tau, \hat{u}) = \Big( \sum_{i=1}^{N} Q_i'(\tau_i, u_i^\star) + V(\tau) - \delta(\tau, u^\star) \Big) - \Big( \sum_{i=1}^{N} Q_i'(\tau_i, \hat{u}_i) + V(\tau) - \delta(\tau, \hat{u}) \Big)$$

$$= \sum_{i=1}^{N} Q_i'(\tau_i, u_i^\star) - \sum_{i=1}^{N} Q_i'(\tau_i, \hat{u}_i) + \delta(\tau, \hat{u}).$$

By (11), the first difference is non-positive, so

$$\mathcal{Q}_{\text{LLM}}(\tau, u^\star) - \mathcal{Q}_{\text{LLM}}(\tau, \hat{u}) \leq \delta(\tau, \hat{u}).$$

Finally, by the assumption $\delta(\tau, u) \leq \varepsilon$ for all $u \in \mathcal{U}(\tau)$, we have $\delta(\tau, \hat{u}) \leq \varepsilon$, which yields

$$\mathcal{Q}_{\text{LLM}}(\tau, u^\star) - \mathcal{Q}_{\text{LLM}}(\tau, \hat{u}) \leq \varepsilon.$$

This concludes the proof. $\square$

*Proof of Corollary 3.2.* Under $\varepsilon = 0$, the assumptions of Theorem 3.1 imply $\delta(\tau, u^\star) = 0$ and $\delta(\tau, u) \leq 0$ for all $u \in \mathcal{U}(\tau)$. Applying Theorem 3.1 gives

$$\mathcal{Q}_{\text{LLM}}(\tau, u^\star) - \mathcal{Q}_{\text{LLM}}(\tau, \hat{u}) \leq 0.$$

On the other hand, since $u^\star \in \arg\max_{u \in \mathcal{U}(\tau)} \mathcal{Q}_{\text{LLM}}(\tau, u)$, we also have

$$\mathcal{Q}_{\text{LLM}}(\tau, u^\star) - \mathcal{Q}_{\text{LLM}}(\tau, \hat{u}) \geq 0.$$

Therefore the difference must be zero, i.e., $\mathcal{Q}_{\text{LLM}}(\tau, \hat{u}) = \mathcal{Q}_{\text{LLM}}(\tau, u^\star)$, which implies $\hat{u} \in \arg\max_{u \in \mathcal{U}(\tau)} \mathcal{Q}_{\text{LLM}}(\tau, u)$. If the maximizer is unique, then $\hat{u} = u^\star$. $\square$

### A.2. Proof of Proposition 4.1

We restate the result with full notation and provide the proof, which is motivated by the work of (Akyürek et al., 2022).

**Proposition A.1** (TRM direction-recovery decomposition (formal)). *Fix an agent $i$, a step $t$, and an action $a$ such that the fitted bias is nonzero, $b_i^\star(\tau_{i,t}, a) \neq 0$. Define the target direction*

$$s_{i,t}^\star(a) := \mathrm{sgn}\big(b_i^\star(\tau_{i,t}, a)\big) \in \{-1, +1\},$$

*where* $\mathrm{sgn}(z) = +1$ *if* $z > 0$ *and* $\mathrm{sgn}(z) = -1$ *if* $z < 0$.

*At step $t$, TRM retrieves a set of entries $\mathcal{E}_{i,t}$ based on the embedding of $\tau_{i,t}$. Each entry $e \in \mathcal{E}_{i,t}$ stores a sparse bias map $e.\mathbf{b}$ over actions. Let $\mathrm{supp}(e.\mathbf{b}) := \{a' : e.\mathbf{b}[a'] \text{ is stored}\}$ and define*

$$\mathcal{E}_{i,t}(a) := \{e \in \mathcal{E}_{i,t} : a \in \mathrm{supp}(e.\mathbf{b})\}.$$

*We say action $a$ is* direction-unambiguous *in $\mathcal{E}_{i,t}$ if $\mathcal{E}_{i,t}(a) \neq \emptyset$ and all $e \in \mathcal{E}_{i,t}(a)$ agree on the sign of $e.\mathbf{b}[a]$. Define the retrieved direction $\tilde{s}_{i,t}(a) \in \{-1, 0, +1\}$ as*

$$\tilde{s}_{i,t}(a) = \begin{cases} \mathrm{sgn}(e.\mathbf{b}[a]), & \text{if } a \text{ is direction-unambiguous in } \mathcal{E}_{i,t} \text{ (for any } e \in \mathcal{E}_{i,t}(a)\text{)}, \\ 0, & \text{otherwise.} \end{cases}$$

*Let $\hat{s}_{i,t}(a) \in \{-1, +1\}$ denote the direction induced by prompting with the retrieved rules. Let $\Pr[\cdot]$ denote probability over retrieval randomness and the LLM's stochastic generation. Assume: (i) (retrieval robustness when unambiguous) $\Pr[\tilde{s}_{i,t}(a) \neq 0 \ \wedge \ \tilde{s}_{i,t}(a) \neq s_{i,t}^\star(a)] \leq \eta_{\mathrm{ret}}$; (ii) (in-context compliance) $\Pr[\hat{s}_{i,t}(a) \neq \tilde{s}_{i,t}(a) \mid \tilde{s}_{i,t}(a) \neq 0, \ \tilde{s}_{i,t}(a) = s_{i,t}^\star(a)] \leq \eta_{\mathrm{comp}}$. Then*

$$\Pr[\hat{s}_{i,t}(a) \neq s_{i,t}^\star(a)] \ \leq \ \Pr[\tilde{s}_{i,t}(a) = 0] \ + \ \eta_{\mathrm{ret}} \ + \ \eta_{\mathrm{comp}}.$$

*Proof of Proposition A.1.* Fix $(i, t, a)$ and abbreviate $s^\star := s_{i,t}^\star(a)$, $\tilde{s} := \tilde{s}_{i,t}(a)$, and $\hat{s} := \hat{s}_{i,t}(a)$. Define the mismatch event

$$\mathcal{A} := \{\hat{s} \neq s^\star\}.$$

Decompose $\mathcal{A}$ according to whether $\tilde{s} = 0$ or not. Let

$$\mathcal{E}_0 := \{\tilde{s} = 0\}, \qquad \mathcal{E}_{\mathrm{ret}} := \{\tilde{s} \neq 0 \ \wedge \ \tilde{s} \neq s^\star\}, \qquad \mathcal{E}_{\mathrm{comp}} := \{\tilde{s} \neq 0 \ \wedge \ \tilde{s} = s^\star \ \wedge \ \hat{s} \neq \tilde{s}\}.$$

We claim that

$$\mathcal{A} \subseteq \mathcal{E}_0 \ \cup \ \mathcal{E}_{\mathrm{ret}} \ \cup \ \mathcal{E}_{\mathrm{comp}}.$$

Indeed, on $\mathcal{A}$ either (i) $\tilde{s} = 0$, which implies $\mathcal{E}_0$; or (ii) $\tilde{s} \neq 0$. Under (ii), either $\tilde{s} \neq s^\star$ (hence $\mathcal{E}_{\mathrm{ret}}$ holds), or $\tilde{s} = s^\star$. If $\tilde{s} = s^\star$ and $\hat{s} \neq s^\star$, then necessarily $\hat{s} \neq \tilde{s}$, so $\mathcal{E}_{\mathrm{comp}}$ holds.

Taking probabilities and applying the union bound yields

$$\Pr[\mathcal{A}] \leq \Pr[\mathcal{E}_0] + \Pr[\mathcal{E}_{\mathrm{ret}}] + \Pr[\mathcal{E}_{\mathrm{comp}}].$$

By Assumption (i),

$$\Pr[\mathcal{E}_{\mathrm{ret}}] = \Pr[\tilde{s} \neq 0 \ \wedge \ \tilde{s} \neq s^\star] \leq \eta_{\mathrm{ret}}.$$

By Assumption (ii),

$$\begin{aligned} \Pr[\mathcal{E}_{\mathrm{comp}}] &= \Pr[\hat{s} \neq \tilde{s} \ \wedge \ \tilde{s} \neq 0 \ \wedge \ \tilde{s} = s^\star] \\ &= \Pr[\hat{s} \neq \tilde{s} \mid \tilde{s} \neq 0, \ \tilde{s} = s^\star] \cdot \Pr[\tilde{s} \neq 0, \ \tilde{s} = s^\star] \\ &\leq \eta_{\mathrm{comp}}. \end{aligned}$$

Combining the bounds gives

$$\Pr[\hat{s} \neq s^\star] \leq \Pr[\tilde{s} = 0] + \eta_{\mathrm{ret}} + \eta_{\mathrm{comp}},$$

as desired. $\square$

---

**Algorithm 1** Residual-Banding and Bias Fitting

---

**Require:** Progress $\tau_t$ and local progress $\{\tau_{i,t}\}_{i=1}^N$; candidate set $\mathcal{U}(\tau_t)$; joint LLM scores $\{\mathcal{Q}_{\text{LLM}}(\tau_t, u)\}_{u \in \mathcal{U}(\tau_t)}$; local LLM scores $\{q_{\text{LLM},i}(\tau_{i,t}, a)\}_{i,a}$; tolerance $\varepsilon_{\text{res}} \geq 0$; max sweeps $S_{\max}$.

**Ensure:** Bias tables $\{b_i(\tau_{i,t}, a)\}_{i,a}$ and baseline $V(\tau_t)$.

 1: Initialize $b_i(\tau_{i,t}, a) \leftarrow 0$ and $Q'_i(\tau_{i,t}, a) \leftarrow q_{\text{LLM},i}(\tau_{i,t}, a)$ for all $i, a$.
 2: Compute joint best candidate: $u^\star \leftarrow \arg\max_{u \in \mathcal{U}(\tau_t)} \mathcal{Q}_{\text{LLM}}(\tau_t, u)$.
 3: Anchor the baseline using $u^\star$ (Eq. (8)): $V(\tau_t) \leftarrow \mathcal{Q}_{\text{LLM}}(\tau_t, u^\star) - \sum_{i=1}^N Q'_i(\tau_{i,t}, u_i^\star)$,
    so that $\delta(\tau_t, u^\star) = 0$.
 4: **for** $s = 1$ to $S_{\max}$ **do**
 5:     VIOLATED $\leftarrow$ **False**.
 6:     **for** each $u = (u_1, \ldots, u_N) \in \mathcal{U}(\tau_t)$ **do**
 7:         Compute residual (Eq. (5)):

$$\delta(\tau_t, u) \leftarrow \sum_{i=1}^N Q'_i(\tau_{i,t}, u_i) + V(\tau_t) - \mathcal{Q}_{\text{LLM}}(\tau_t, u).$$

 8:         **if** $|\delta(\tau_t, u)| > \varepsilon_{\text{res}}$ **then**
 9:             VIOLATED $\leftarrow$ **True**.
10:             $e(\tau_t, u) \leftarrow \delta(\tau_t, u) - \text{clip}(\delta(\tau_t, u), -\varepsilon_{\text{res}}, \varepsilon_{\text{res}})$.
11:             $\alpha \leftarrow -e(\tau_t, u)/N$ {uniform correction share}
12:             **for** $i = 1$ to $N$ **do**
13:                 $b_i(\tau_{i,t}, u_i) \leftarrow b_i(\tau_{i,t}, u_i) + \alpha$
14:                 $Q'_i(\tau_{i,t}, u_i) \leftarrow q_{\text{LLM},i}(\tau_{i,t}, u_i) + b_i(\tau_{i,t}, u_i)$
15:             **end for**
16:         **end if**
17:     **end for**
18:     Re-anchor $V(\tau_t)$ to keep $\delta(\tau_t, u^\star) = 0$:

$$V(\tau_t) \leftarrow \mathcal{Q}_{\text{LLM}}(\tau_t, u^\star) - \sum_{i=1}^N Q'_i(\tau_{i,t}, u_i^\star).$$

19:     **if not** VIOLATED **then**
20:         **break** {All candidates satisfy $|\delta(\tau_t, u)| \leq \varepsilon_{\text{res}}$.}
21:     **end if**
22: **end for**
23: **return** $\{b_i(\tau_{i,t}, a)\}_{i,a}$ and $V(\tau_t)$.

---

# B. Algorithm

The algorithms for Residual banding and bias fitting is provided in Algorithm 1

# C. Additional Results

## C.1. Case Study

Figure 3 illustrates a representative coordination failure under partial observability, and how *Tact* resolves it via retrieved rules. The task is to place an INGREDIENT (apple) onto a TARGET (wooden cutting board) and then slice it using a TOOL (knife). With the vanilla decentralized baseline (P-dec), each robot plans independently from its local observation and implicitly assumes it should pursue the primary object. As a result, *both* robots choose the same action sequence (Move/Reach/Grasp apple followed by Place apple $\rightarrow$ board), producing a redundant and conflicting joint plan. This joint plan is ill-posed in the environment because the two robots compete for the same object (duplicate grasp/place), and no agent reliably commits to fetching and using the knife, preventing completion of the slicing subgoal.

Figure 3. **Case study.** A decentralized planner (left) fails under partial observability: both robots independently plan to manipulate the same ingredient (apple), resulting in redundant/conflicting actions (duplicate grasp/place) and missing the tool-usage subgoal. With *Tact* (right), each robot still plans locally but conditions on retrieved TRM rules, yielding an implicit division of labor (R1 handles INGREDIENT→TARGET, R2 handles TOOL and performs the final Interact) and a consistent joint plan without explicit messaging.

In contrast, *Tact* retrieves complementary TRM rules that impose an implicit division of labor and a shared ordering over subgoals. In this example, R1 is assigned an *ingredient-chain* role (collect INGREDIENT → go to TARGET → place INGREDIENT) and is explicitly forbidden from touching the tool, while R2 is assigned a *tool-chain* role (collect TOOL → go to TARGET → Interact(TOOL)) and is forbidden from manipulating the ingredient. Conditioning each local planner on these rules yields coordinated, non-overlapping actions: R1 manipulates only the apple and places it at the cutting board, while R2 retrieves the knife, waits at the board if necessary, and performs the final Interact(knife) as the last step. This case highlights the core benefit of *Tact*: without any online messaging, retrieved rule context provides a tacit communication signal that prevents mutual-exclusion violations and enforces correct subtask sequencing.

Importantly, the Do/Don't directives shown in Figure 3 are *not* produced by free-form rule writing. They are grounded in the centralized bias fitting stage: we identify actions with the largest-magnitude fitted biases $b_i^\star(\tau_i, a)$ (strongly encouraged vs. strongly discouraged under the current local progress) and treat them as *action-grounded supervision*. Concretely, for each agent we construct an 'encourage' set and a 'discourage' set. We then prompt an LLM to summarize these grounded action lists into a compact cooperation rule with a fixed schema (Role, Do, Don't, Ordering). Thus, TRM rules are distilled from residual-aligned bias signals and inherit strong, decision-relevant supervision, rather than relying on unconstrained natural-language generation.

## C.2. Additional results on TDW-MAT

We evaluate TDW-MAT in a setting where agents *do not* have access to oracle instance segmentation. Following prior work (Zhang et al., 2023), we instantiate the perception module with a Mask R-CNN detector trained on collected TDW scene images. The detector outputs object instances (targets and containers), which are converted into text-based observations (object category/name, estimated location cues, and held-object states) that are consumed by the LLM-based planners. We report Transport Rate (%).

Table 7 summarizes results with two-agent teams. Under imperfect perception, the decentralized baseline methods are prone

*Table 7.* Transport rate (%, ↑) on TDW-MAT without oracle perception. Higher is better. RHP* uses a single agent, while all others adopt two agents.

| | RHP | | Qwen2.5-72B-Instruct | | | | GPT-4o | | | |
|---|---|---|---|---|---|---|---|---|---|---|
| | RHP* | RHP | CoELA | ProAgent | RoCo | *Tact* | CoELA | ProAgent | RoCo | *Tact* |
| Food (↑) | 49.0 | 67.0 | 77.0 | 73.2 | 72.3 | 73.4 | 81.1 | 75.5 | 77.2 | 79.6 |
| Stuff (↑) | 36.0 | 54.0 | 68.1 | 63.4 | 65.0 | 68.3 | 74.2 | 74.0 | 74.9 | 76.0 |
| Avg. (↑) | 42.5 | 60.5 | 72.6 | 68.3 | 68.7 | 70.9 | 77.7 | 74.8 | 76.1 | 77.8 |

*Table 8.* Direction-recovery on the held-out split. Lower is better.

| Method | $\text{ACC}_{\text{OOD}} \uparrow$ | $\Pr[\hat{s} \neq s^\star] \downarrow$ | $\Pr[E_{\text{gap}}] \downarrow$ | $\Pr[E_{\text{dir}}] \downarrow$ | $\Pr[E_{\text{comp}}] \downarrow$ |
|---|---|---|---|---|---|
| Full TRM | 30.2 | 0.28 | 0.22 | 0.08 | 0.06 |
| w/o bias sup | 1.2 | 0.75 | 0.55 | 0.30 | 0.04 |
| Random retrieval / no useful TRM | 1.8 | 0.68 | 0.80 | 0.06 | 0.02 |

to redundant exploration and conflicting object choices, which is amplified when multiple targets and containers must be coordinated across rooms. Across both backbones, *Tact* achieves the best Stuff transport rate. On Food transport, *Tact* remains competitive.

### C.3. Additional Ablations

We conduct a small *direction-recovery audit* that directly instantiates Proposition 4.1. For each sampled decision point on the held-out split, we treat the fitted bias direction $s^\star(a) \in \{-1, +1\}$ (encourage vs. discourage) as the target signal for each action $a$. We run TRM retrieval to obtain a retrieved direction $\tilde{s}(a) \in \{-1, 0, +1\}$, where $\tilde{s}(a) = 0$ indicates that the retrieved entries provide no *consistent* directional preference for $a$ (missing coverage or conflicting signs). We then prompt the agent with the retrieved rules and record the induced direction $\hat{s}(a) \in \{-1, +1\}$. Following Proposition 4.1, we estimate the mismatch rate $\Pr[\hat{s} \neq s^\star]$ and its decomposition into three events: $E_{\text{gap}}$ (no consistent retrieved signal), $E_{\text{dir}}$ (consistent-but-wrong retrieved direction), and $E_{\text{comp}}$ (non-compliance to a correct unambiguous direction). Lower is better for all probabilities.

Table 8 shows that Full TRM achieves the lowest direction mismatch (0.28) and a substantially smaller transfer gap ($\Pr[E_{\text{gap}}] = 0.22$). Removing bias supervision sharply increases mismatch (0.75) and, importantly, yields a large consistent-but-wrong term ($\Pr[E_{\text{dir}}] = 0.30$), indicating that non-action-grounded rules can become *misleading* on the held-out split and cause OOD collapse. In contrast, random retrieval produces an extreme gap ($\Pr[E_{\text{gap}}] = 0.80$) but a small wrong-direction term ($\Pr[E_{\text{dir}}] = 0.06$), because random memories usually fail by providing no consistent signal. Across settings, $\Pr[E_{\text{comp}}]$ remains small, suggesting that when a correct unambiguous direction is retrieved, the agent largely follows it; the main bottleneck is therefore producing *covered and non-conflicting* directional guidance (reducing $E_{\text{gap}}$), which is what TRM with bias-grounded rule construction provides.

*Table 9.* LLM critic reliability and robustness on VIKI-L2. $Q_{\text{gt}} > Q_{\text{local}}$ measures the fraction of intermediate states where the critic ranks the expert action no lower than the local alternative.

| Critic Variant | $Q_{\text{gt}} > Q_{\text{local}}$ (%) ↑ | $\text{ACC}_{\text{ID}} \uparrow$ |
|---|---|---|
| Pseudo-Oracle | 100.0 | 91.3 |
| LLM Critic (proposed) | 87.5 | 89.2 |
| Noisy LLM ($\sigma = 0.5$) | 76.8 | 78.2 |
| Noisy LLM ($\sigma = 1.0$) | 64.6 | 52.4 |
| Random Critic | 48.7 | 4.3 |

We further analyze whether the intermediate LLM critic provides meaningful task-relevant value signals. We compare the proposed LLM critic with three alternatives: (i) a pseudo-oracle critic constructed from ground-truth labels, which serves as an upper-bound scoring signal; (ii) noisy variants of the LLM critic, where we perturb the critic score as $\tilde{\mathcal{Q}}_{\text{LLM}}(\tau_t, u) = \mathcal{Q}_{\text{LLM}}(\tau_t, u) + \epsilon$ with $\epsilon \sim \mathcal{N}(0, \sigma^2)$; and (iii) a random critic. We report $Q_{\text{gt}} > Q_{\text{local}}$, the fraction of intermediate states where the critic scores the expert action no lower than the local alternative, together with downstream $\text{ACC}_{\text{ID}}$.

*Table 10.* VIKI robot embodiments and per-robot available APIs (used in VIKI-L2 planning).

| Type | Description | Mobile | Allowed APIs |
|------|-------------|--------|--------------|
| stompy | Bipedal robot with articulated arms; suited for manipulation. | Yes | Move, Reach, Grasp, Place, Open, Close, Interact |
| fetch | Wheeled mobile manipulator with a flexible arm. | Yes | Move, Reach, Grasp, Place, Open, Close, Interact |
| unitree_h1 | Humanoid robot with arms and legs for human-like tasks. | Yes | Move, Reach, Grasp, Place, Open, Close, Interact |
| panda | Fixed robotic arm for precise manipulation. | No | Reach, Grasp, Place, Open, Close, Interact |
| anymal_c | Quadruped; locomotion-focused (no end effector). | Yes | Move, Push, Interact |
| unitree_go2 | Compact quadruped; locomotion-focused (no end effector). | Yes | Move, Push, Interact |

*Table 11.* VIKI action primitives and brief semantics (used in VIKI-L2).

| Primitive | Description |
|-----------|-------------|
| Move | Move to the specified object/location. |
| Reach | Reach toward the specified object. |
| Grasp | Execute a grasp on the specified object (requires an end effector). |
| Place | Place the held object at a specified release location. |
| Open | Open the specified object (e.g., a container/door) using the end effector. |
| Close | Close the specified object using the end effector. |
| Push | Push an object toward a target robot (e.g., Push(object, R1)). |
| Interact | A general interaction primitive for task-specific interactions. |

As shown in Table 9, the proposed LLM critic ranks the expert action above the local alternative in $87.5\%$ of intermediate states, substantially higher than the random critic. Downstream performance also tracks critic quality: moderate noise causes a degradation, while a random critic nearly collapses performance. These results indicate that the intermediate LLM critic carries task-relevant information rather than merely reflecting linguistic patterns.

# D. Details of Environments and Implementations

## D.1. Environmental Details

### D.1.1. VIKI-L2 (VIKI-BENCH: TASK PLANNING)

VIKI-L2 is the *task planning* level of VIKI-Bench, where the model generates a structured multi-robot plan from a global-view scene image, a task description, and the available robot set with their action primitives, which is provided in table 10 and table 11. The output is a time-indexed plan: at each step, each robot may take at most one primitive action, and multiple robots may act in parallel. The benchmark uses heterogeneous robot embodiments and evaluates whether a predicted plan is feasible (and, in the original benchmark protocol, not longer than the reference plan). We report results on both in-domain (ID) and held-out out-of-domain (OOD) splits. A VIKI-L2 plan is a list of steps; each step contains an actions dictionary mapping robot IDs (e.g., R1, R2) to a single primitive action tuple (primitive name, target object/location, optional argument).

### D.1.2. C-WAH (COMMUNICATIVE WATCH-AND-HELP)

C-WAH extends the Watch-And-Help household benchmark with multi-agent messaging. Each episode specifies a set of symbolic subgoals as predicates of the form ON/IN(x, y) (e.g., "put x IN y"). The objective is to complete all specified predicates within a fixed horizon (250 timesteps), typically 3–5 subtasks per episode.

*Table 12.* C-WAH task families and predicate templates.

| Task family | Predicate set (examples) |
|---|---|
| Prepare afternoon tea | `ON(cupcake, coffeetable)`, `ON(pudding, coffeetable)`, `ON(apple, coffeetable)`,`ON(juice, coffeetable)`,`ON(wine, coffeetable)` |
| Wash dishes | `IN(plate, dishwasher)`,`IN(fork, dishwasher)` |
| Prepare a meal | `ON(coffeepot, dinnertable)`, `ON(cupcake, dinnertable)`, `ON(pancake, dinnertable)`, `ON(poundcake, dinnertable)`, `ON(pudding, dinnertable)`, `ON(apple, dinnertable)`, `ON(juice, dinnertable)`, `ON(wine, dinnertable)` |
| Put groceries | `IN(cupcake, fridge)`, `IN(pancake, fridge)`, `IN(poundcake, fridge)`, `IN(pudding, fridge)`, `IN(apple, fridge)`, `IN(juice, fridge)`, `IN(wine, fridge)` |
| Set up a dinner table | `ON(plate, dinnertable)`,`ON(fork, dinnertable)` |

*Table 13.* C-WAH observation and action spaces.

| Component | Details |
|---|---|
| Symbolic obs. | Room-level symbolic state with object location/status/name and relations. |
| Visual obs. | Egocentric RGB ($256\times512$, FOV $60°$), depth, optional oracle perception, agent position, messages, held objects, opponent-held objects if visible. |
| Actions | `Walk towards`, `Turn left/right` ($30°$), `Grasp`, `Open`, `Close`, `Put`, `Send message` (length-limited). |
| Horizon | 250 timesteps per episode. |

**Observation space.** C-WAH supports two observation settings: **symbolic** (room-level symbolic state with object attributes/relations) and **visual** (egocentric RGB-D plus auxiliary state such as positions, held objects, and messages). As provided in table 12.

**Action space.** Agents can navigate and interact via discrete high-level actions (e.g., walk toward, grasp, open/close, put), with an additional `send message` action. As provided in table 13.

### D.1.3. TDW-MAT (THREEDWORLD MULTI-AGENT TRANSPORT)

TDW-MAT is a two-agent transport benchmark in the TDW simulator. Each episode contains multiple target objects distributed across rooms, and the goal is to transport as many targets as possible to the goal location (bed) within a 3000-frame budget. The environment includes **containers** that can hold up to three objects, enabling more efficient transport than carrying objects by hand.

**Task types.** TDW-MAT includes **food transport** and **stuff transport**. Each episode contains 10 target objects and 2–5 containers across four room types (living room, office, kitchen, bedroom). As shown in table 14.

**Observation space.** Agents receive egocentric RGB-D (and optional oracle perception) with auxiliary state such as positions, held objects, and (optional) teammate-held objects if visible.

**Action space.** Agents act with discrete navigation and manipulation actions; each action may span multiple frames. Messaging is supported as a discrete action with a per-frame character limit. As shown in table 15.

### D.2. Implementation Details

#### D.2.1. EXPERIMENTAL SETUP.

For closed-source backbones, we access GPT-4o via the OpenAI API. For open-source backbones, we deploy Qwen models from HuggingFace locally. All local inference is run on workstations equipped with NVIDIA RTX PRO 6000 Blackwell and NVIDIA H200 GPUs.

We follow the standard evaluation protocol used in prior work on TDW-MAT and C-WAH. In TDW-MAT, the official split

*Table 14.* TDW-MAT task specification and constraints.

| Item | Details |
|---|---|
| Food targets | `apple, banana, orange, bread, loaf bread, burger` |
| Food containers | `bowl, plate, tea tray` |
| Stuff targets | `calculator, mouse, pen, lighter, purse, iPhone` |
| Stuff containers | `plastic basket, wood basket, wicker basket` |
| Rooms | `living room, office, kitchen, bedroom` |
| Capacity rule | Each container holds up to 3 objects; without a container an agent carries up to 2 objects. |
| Objective / horizon | Transport as many targets as possible to the goal (bed) within 3000 frames. |

*Table 15.* TDW-MAT observation and action spaces.

| Component | Details |
|---|---|
| Observations | Egocentric RGB ($512 \times 512$, FOV $90°$), depth, optional oracle perception, agent position/rotation, messages, held objects, opponent-held objects if within view. |
| Actions | `Move forward` (0.5m), `Turn left/right` ($15°$), `Grasp` (target or container), `Put In` (target into held container), `Drop`, `Send message` (length-limited). |

separates training and test floorplans (two floorplans for training and two for testing) (Zhang et al., 2023), and the reported test set contains 24 episodes constructed from 6 scenes with both food and stuff tasks. In C-WAH, the reported test set contains 10 episodes (2 tasks sampled from each of the 5 activity types, horizon 250).

All *bias fitting* and TRM rule distillation are performed *offline* on the training split only. During this phase, we may query a centralized joint LLM evaluator to score candidate joint plans and fit per-agent biases. At *evaluation* time, we disable explicit message actions and freeze TRM; each agent acts purely from its local observation/history and the retrieved rule texts, and we do not query the centralized evaluator or update biases/rules online. Therefore, the centralized scorer is only an offline distillation tool and is not part of decentralized execution.

D.2.2. IMPLEMENTATION DETAILS OF THE LLM CRITIC AND LOG-ODDS VALUES.

In our implementation, we do not rely on token-level log-probabilities from proprietary chat APIs. Instead, all *bias fitting* and offline rule mining are performed with a local HuggingFace model, for which we can compute *log-odds* critic value $Q_{LLM}(\tau, u) = \log P(y = \texttt{success} \mid \tau, u) - \log P(y = \texttt{fail} \mid \tau, u)$ exactly.

Concretely, given a prompt that describes the state and a candidate (joint or local) action, we append a one-token continuation 'success' or 'fail' and compute the summed token log-likelihood of the continuation under the local causal LM. For a continuation string $c$ and prompt $x$, we compute $\log P(c \mid x) = \sum_{t=1}^{|c|} \log P(c_t \mid x, c_{<t})$ via the model's logits and a log-softmax, and then form the log-odds as $\log P(\texttt{success} \mid x) - \log P(\texttt{fail} \mid x)$. We run the critic deterministically (temperature $= 0$) to reduce variance. This procedure is implemented in our codebase as `_hf_logprob_of_continuation()` and used by `global_critic()` and `local_policy_logits()`.

At deployment time, the distributed execution stage does not require log-probabilities: it retrieves the distilled rules/context from the rule memory (TRM) and conditions the planner on these retrieved snippets. Therefore, even though closed-source models (e.g., OpenAI GPT-4o) may not expose token-level log-probabilities, this does not affect our decentralized execution pipeline. In short, log-odds are computed only offline using an open-source local model for bias fitting and rule mining, while the online/distributed stage only performs retrieval and rule-conditioned planning.

D.2.3. CONSTRUCTION OF THE JOINT CANDIDATE SET $\mathcal{U}(\tau_t)$ AND MITIGATING COMBINATORIAL BLOW-UP.

A natural concern is that enumerating joint candidates scales as $\prod_i |\mathcal{A}_{i,t}|$, which becomes intractable for $N \geq 3$ agents. In our implementation we avoid the full Cartesian product and instead construct a *small structured subset* $\mathcal{U}(\tau_t)$, tailored to the evaluation setting, such that centralized scoring and residual fitting remain efficient.

In CWAH and TDW-MAT, the optimal joint action is not available and dynamics are stochastic, so we cannot anchor $\mathcal{U}(\tau_t)$ on ground-truth optimal actions. Instead, at each step we take the executed joint action $u^{\text{exec}} = (a_1^{\text{exec}}, \ldots, a_N^{\text{exec}})$ from the logged trajectory and build *one-sided counterfactual* candidates by varying one agent at a time while holding the others

fixed. Concretely, for each agent $i$ we compute

$$u_i^{\text{cf}} = \arg \max_{a_i \in \mathcal{A}_{i,t}} \mathcal{Q}_{\text{LLM}}(\tau_t, (a_1^{\text{exec}}, \ldots, a_i, \ldots, a_N^{\text{exec}})), \tag{12}$$

and set

$$\mathcal{U}(\tau_t) = \{u^{\text{exec}}\} \cup \{u_i^{\text{cf}}\}_{i=1}^N, \tag{13}$$

deduplicated. This construction requires only $\sum_i |\mathcal{A}_{i,t}|$ centralized critic calls per step (linear in the number of per-agent candidates), rather than $\prod_i |\mathcal{A}_{i,t}|$. Intuitively, these one-sided counterfactuals capture the most informative deviations for diagnosing coordination failures and fitting per-agent residual biases, while keeping the joint evaluation budget bounded.

In VIKI, we have access to the logged optimal joint action from ground-truth supervision, denoted $u^\star$ (the GT action at the current step). We therefore explicitly include $u^\star$ in $\mathcal{U}(\tau_t)$ as an anchor, together with a small set of decentralized alternatives constructed from local proposals. Specifically, for each agent $i$ we obtain a local greedy proposal $a_i^{\text{loc}}$, form the joint proposal $u^{\text{loc}} = (a_1^{\text{loc}}, \ldots, a_N^{\text{loc}})$, and optionally include single-agent deviations

$$u_i^{\text{dev}} = (u_1^\star, \ldots, a_i^{\text{loc}}, \ldots, u_N^\star). \tag{14}$$

Our final candidate set is

$$\mathcal{U}(\tau_t) = \{u^\star, u^{\text{loc}}\} \cup \{u_i^{\text{dev}}\}_{i=1}^N, \tag{15}$$

again deduplicated. This yields $|\mathcal{U}(\tau_t)| \leq 2 + N$, making the centralized scoring cost essentially linear in $N$ and independent of $\prod_i |\mathcal{A}_{i,t}|$. Crucially, including $u^\star$ ensures that residual fitting and rule mining are always grounded in the known optimal action in this offline-supervised setting.

Across both settings, we avoid combinatorial explosion by constructing $\mathcal{U}(\tau_t)$ as a small, structured subset composed of (i) an anchor joint action ($u^{\text{exec}}$ for stochastic environments; $u^\star$ for VIKI), and (ii) a handful of one-agent counterfactual/deviation joints. This design keeps centralized evaluation and residual fitting scalable while retaining the most decision-relevant joint alternatives for coordination diagnostics and rule extraction.

### D.2.4. TRM IMPLEMENTATION DETAILS.

TRM stores distilled coordination rules as `RuleRecords`, each containing a retrieval key `key_text` (task description + robot id + an observation snippet) and per-robot rule fields (e.g., `role`, `ordering`, `do/don't`). At query time we build the same `key_text` for the current episode and retrieve relevant rules.

*Encoder and similarity.* We use a lightweight local text encoder and compute embeddings by mean-pooling the last-layer hidden states (mask-aware). Similarity is cosine: $s(q, x) = \langle e(q), e(x) \rangle / (\|e(q)\| \|e(x)\|)$. Embeddings for the memory are precomputed offline and stored and query embeddings are computed online both in float16.

*Retrieval.* For scalability, retrieval is two-stage: (i) TF–IDF coarse recall over `key_text` selects $K_1{=}20$ candidates, (ii) cosine reranking over embeddings returns top $K_2{=}5$. We then keep the top-$k$ episodes (default $k{=}1$).

*Conflict handling and budget.* We avoid rule conflicts and prompt bloat by using small $k$ and rank-based precedence: retrieved rules are injected in decreasing similarity order, and we only render supervised rule summaries (role/ordering/do), keeping the injected rule context bounded and stable.

### D.2.5. DECENTRALIZED VARIANT OF VIKI-L2.

To study *decentralized* multi-robot planning under partial observability, we modify VIKI-L2 only at the *planner interface* level (the environment, goal constraints, and evaluator remain unchanged). At each timestep $t$, instead of providing a centralized planner with a joint decision variable $u_t = (u_t^1, \ldots, u_t^N)$, we run an *independent* planner for each robot $i$ that outputs only its own action $u_t^i$. Concretely, the input to robot $i$ is a local prompt

$$p_t^i = \left[ d \parallel o \parallel (i, \text{type}_i) \parallel \mathcal{A}_i \parallel h_{<t}^i \right],$$

where $d$ is the task description, $o$ is a natural-language scene summary, $\mathcal{A}_i$ is the robot-specific action API set parsed from the original VIKI prompt, and $h_{<t}^i$ is *only robot $i$'s own executed action history* (no teammate actions or intentions are exposed). The decentralized baseline runs these per-robot planners in parallel and merges the resulting per-robot plans into a joint plan

*Table 16.* Prompt template for rule text generation on VIKI-L2.

```
You are a tacit rule author.  You write clear, high-impact coordination rules that
enable decentralized teamwork WITHOUT explicit messaging.
You must infer this robot's responsibilities by generalizing ONLY from the DO and
DON'T action evidence below; do not invent duties beyond the evidence.
Robot:  {rid} ({rtype})
Task:  {task_desc}
{global_summary}
=== DO ACTION EVIDENCE (what YOU should prioritize) ===
{do_chain}
=== DON'T ACTION EVIDENCE (what YOU should avoid to reduce conflicts or redundancy)
===
{other_block}
Generalization vocabulary (for transfer across scenes):
When possible, express responsibilities using abstract object categories such as:
- TOOL
- INGREDIENT
- TARGET
- CONTAINER
CRITICAL REQUIREMENTS:
- Output STRICT JSON only (no additional text), with keys:  "role", "do", "dont",
"ordering".
- "role":  one phrase describing this robot's unique responsibility in the team.
- "do":  1--4 generalized responsibilities implied by YOUR DO evidence.
- "dont":  1--4 generalized prohibitions implied by YOUR DON'T evidence.
- "ordering":  a phase chain using category words (e.g., "TOOL -> INGREDIENT ->
TARGET -> CONTAINER"), consistent with "do".
Return STRICT JSON only:
{
  "role":  "...",
  "do":  ["...", "..."],
  "dont":  ["...", "..."],
  "ordering":  "phase1 -> phase2 -> phase3"
}
```

by aligning timesteps (`merge_multi_robot_plans()`); the centralized baseline uses a separate joint-planning prompt that outputs all robots' actions directly.

*Why this matches decentralized embodied coordination and motivates TRM.* This local-observation interface reflects a realistic decentralized setting where a robot does not have access to teammates' internal states, private reasoning, or intended future actions; it only observes the scene and its own action history. Under such partial observability, coordination failures often arise from *ambiguous teammate intent*. TRM addresses this by retrieving a small number of relevant coordination rules and injecting them into each robot's local prompt, providing a shared *tacit* convention for role assignment and action ordering without any online communication.

# E. Prompt Template

*Table 17.* Prompt template for rule-conditioned decentralized planning on VIKI-L2.

```
You are a plan creator.  I will provide you with the current observation, available
robots, and a task description.  You need to create a plan to complete the task.
Analyze the observation, the task description, and the valid action APIs before
producing the plan.
Your plan must strictly adhere to the observation and the task description---no
assumptions, hypotheses, or guesses are allowed.

Tacit cooperation setting (no explicit messaging):
You are planning ONLY for {robot_id}, which is a {robot_type}.  Other robots may also
act in parallel.
Follow the retrieved rules below to coordinate implicitly, avoid conflicts, and
prevent redundant work (e.g., two robots grasping the same object at the same time).
If an action should be avoided according to the retrieved rules or division of labor,
do NOT output it.

Retrieved Cooperation Rules:
{retrieved_rules}

Hard constraints:
1) Each robot can only perform ONE action per time step.
2) Multiple robots can work in parallel, but each robot is limited to one action at
a time.
3) Output actions must be valid for THIS robot type {robot_type} and its allowed
APIs.

Output requirements:
1) Your final answer must be within <answer> and </answer> tags, and strictly follow
the JSON format below.
2) Output ONLY the JSON inside <answer> tags; do not include extra text.

Output Format:
<answer>
[
  {
    "step":  1,
    "actions":  {"{robot_id}":  ["Move", "banana"]}
  },
  {
    "step":  2,
    "actions":  {"{robot_id}":  ["Reach", "banana"]}
  }
]
</answer>

Inputs:
Observation:  {observation}
Task:  {task_desc}
Action primitives and descriptions:  {action_primitives}
Available robot set:  {available_robots}
Allowed operation APIs for THIS robot ({robot_type}):  {robot_allowed_apis}
Their available operation APIs (all robot types):  {robot_apis}
```

