# OpenReview forum: "Great Minds Think Alike: Contextual Tacit Communication for Decentralized LLM-Agent Cooperation"
_ICML.cc/2026/Conference — ICML 2026 regular_

### Official Review · Reviewer_zJkG · 2026-03-11

**Soundness:** 2
**Presentation:** 3
**Significance:** 2
**Originality:** 2
**Overall Recommendation:** 4
**Confidence:** 3

**Summary:**

This paper proposes Tact, a training-free tacit communication protocol for decentralized LLM-agent cooperation that aligns local and global LLM value scores via residual banding and distills coordination signals into a retrieval-augmented rule memory for implicit guidance, achieving strong cooperative performance and lower inference overhead on three embodied multi-agent benchmarks with theoretical near-optimality guarantees.

**Compliance With Llm Reviewing Policy:**

Affirmed.

**Final Justification:**

The author resolved my questions, so I decided to raise my rating to 4.

**Key Questions For Authors:**

see above.

**Limitations:**

yes

**Strengths And Weaknesses:**

**Strengths**
1. This paper addresses a meaningful and promising research direction in multi-agent collaboration, focusing on tacit communication for decentralized LLM agents to mitigate the limitations of explicit communication.
2. This paper is well-structured with a clear and logical narrative throughout the methodology, experiments and analysis sections.
3. The theoretical underpinnings of the proposed method are thoroughly elaborated and rigorously proven with formal definitions, theorems and corollaries.

**Weaknesses**
1. The proposed method relies heavily on LLM correctness. What is the performance sensitivity when LLM scoring is noisy or inaccurate?  In particular, the unreliability of LLM's "ground truth" in more complex environments is likely to result in a significant degradation of the overall method's performance.
2. While Tact presents a reasonable design, its overall methodological novelty is limited. Residual banding and TRM are largely built upon existing techniques from value decomposition and retrieval-augmented prompting, without introducing fundamental new principles.
3. When multiple optimal joint actions exist, how does Tact ensure that the learned bias values and TRM rules are consistent and conflict-free? The paper does not discuss the behavior or stability of the method under such scenarios.
4. Theorem 3.1 assumes that the shared baseline $V(\tau_t)$ is independent of the joint action, which appears inconsistent with its definition. As $V(\tau_t)$ is constructed from the optimal joint action, it seems inherently dependent on the action set. Please clarify the validity of this assumption and the associated theoretical guarantee.

---

> ### Author Rebuttal · Authors · 2026-03-31
>
> We thank the reviewer for the thoughtful review and for recognizing our work as addressing a meaningful research direction. We respond to each weakness below.
>
> **W.1 Sensitivity to LLM critic noise.**
>
> We added a critic reliability and robustness analysis to directly evaluate this concern:
>
> **Table a4. LLM Critic Reliability and Robustness on VIKI.** We compare: (i) a pseudo-oracle critic constructed from ground thuth labels as an upper-bound scoring signal, (ii) the proposed LLM critic, (iii) noisy LLM critics, (iv) a random critic. We perturb the critic score as $\tilde{Q}_{\mathrm{LLM}}(\tau_t,u)=Q_{\mathrm{LLM}}(\tau_t,u)+\epsilon,\ \epsilon\sim\mathcal{N}(0,\sigma^2)$. *Q_gt > Q_local (%)*: fraction of intermediate states where the critic scores the expert action no lower than the local alternative.
>
> | Critic Variant | Q_gt > Q_local (%) ↑ | ACC_ID ↑ |
> |---|---:|---:|
> | Pseudo-Oracle | 100.0 | 91.3 |
> | **LLM Critic (proposed)** | **87.5** | **89.2** |
> | Noisy LLM (σ=0.5) | 76.8 | 78.2 |
> | Noisy LLM (σ=1.0) | 64.6 | 52.4 |
> | Random Critic | 48.7 | 4.3 |
>
> Key findings: (1) The proposed critic ranks the expert action above the local alternative in 87.5% of intermediate states (vs. 48.7% random), confirming the signal is task-relevant. (2) Performance degrades smoothly under increasing noise rather than catastrophically—at moderate corruption (σ=0.5), most coordination benefit is preserved. (3) Downstream accuracy tracks critic quality monotonically, confirming that Tact's effectiveness is grounded in meaningful critic signals. These results indicate that our method is sensitive to critic quality, but **remains reasonably robust**.
>
>
> **W.2 Methodological novelty.**
>
> The novelty of Tact is not in an isolated retrieval module or value-decomposition component, but in the **protocol-level formulation** connecting them: centralized preferences → action-level coordination supervision (residual banding) → reusable tacit context for decentralized execution (TRM).
>
> Residual banding is not a generic value decomposition adaptation: it formalizes the coordination gap between centralized joint LLM scoring and decentralized greedy selection as a contextual residual, and **Theorem 3.1** links this directly to greedy near-optimality. TRM is not standard retrieval prompting: it distills bias-grounded coordination signals into reusable tacit context, and **Proposition 4.1** formalizes when retrieved rules preserve the desired coordination direction.
>
> To our knowledge, prior work has not formulated tacit coordination for black-box LLM agents in this way—connecting centralized LLM scoring to decentralized execution through residual-aligned rule distillation.
>
> **W.3 Multiple optimal joint actions.**
>
> Our theory does **not** require a unique joint maximizer. Definition 2.1 only requires a non-empty intersection between the centralized optimal set and the decentralized greedy set. **Corollary 3.2** states that under zero residual, the greedy solution belongs to the set of maximizers; uniqueness is needed only for the stronger identity $\hat{u}=u^\star$.
>
> In implementation, residual fitting anchors on a deterministically selected $u^\star$ (enforcing $\delta(\tau,u^\star)=0$), so the learned decomposition is internally consistent with that anchor. For TRM, when retrieved entries disagree in sign, **Proposition 4.1** sets $\tilde{s}(a)=0$ (the $E_{\text{gap}}$ case), and the implementation further reduces conflicts via small top-$k$ retrieval ($k$=1 by default) with rank-based precedence (Appendix D.2.5). In practice, exact ties in $\mathcal{Q}_{\text{LLM}}$ were rare and we did not observe instability.
>
> **W.4 $V(\tau)$ dependence on the joint action.**
>
> We respectfully clarify that this is a misunderstanding of the role of $V(\tau)$ in the theorem.
>
> $V(\tau_t)$ is defined by Eq. (8). For a fixed progress $\tau_t$, the anchor $u^\star = \arg\max_{u \in \mathcal{U}(\tau_t)} \mathcal{Q}_{\mathrm{LLM}}(\tau_t, u)$ is deterministically fixed, and the current rectified values $Q_i'$ are also fixed. Therefore, $V(\tau_t)$ is a **fixed scalar** — it is fully determined once $\tau_t$ is given.
>
> The key statement in the proof of **Theorem 3.1 (Appendix A.1)**, "Since $V(\tau)$ does not depend on $u$," refers to the fact that $V$ **does not vary with the candidate $u$ being evaluated**. That is, when computing the residual $\delta(\tau, u) = \sum_i Q_i'(\tau_i, u_i) + V(\tau) - \mathcal{Q}_{\mathrm{LLM}}(\tau, u)$ for different candidates $u \in \mathcal{U}(\tau)$, the same constant $V(\tau)$ is used throughout. $V$ depends on the anchor $u^\star$ and the current $Q_i'$ values, but these are all fixed quantities — not functions of the comparison variable $u$.
>
> This is fully consistent with the theorem's validity: the proof only requires that the same scalar offset is added to all candidates, so that $\arg\max_u \sum_i Q_i'(\tau_i, u_i)$ is unaffected by $V(\tau)$, and the $\varepsilon$-near-optimality guarantee follows directly.

---

> > ### Author Rebuttal · Reviewer_zJkG · 2026-04-01
> >
> > Thank you for the authors’ detailed response. I will raise my score to 4.

---

> > > ### Author Response · Authors · 2026-04-03
> > >
> > > We thank the reviewer for the thoughtful feedback, for acknowledging the value of our work, and for raising the score. We also greatly appreciate the time and effort invested in reviewing our paper.

---

### Official Review · Reviewer_i9im · 2026-03-12

**Soundness:** 3
**Presentation:** 2
**Significance:** 2
**Originality:** 2
**Overall Recommendation:** 4
**Confidence:** 3

**Summary:**

This paper proposes a training-free, decentralized coordination protocol for closed-source LLM multi-agent systems, introducing the concept of "context tacit communication." Since closed-source LLMs do not allow gradient-based parameter updates, the conventional centralized training with decentralized execution (CTDE) paradigm—which requires a differentiable critic—cannot be directly applied. To address this, the authors use the LLM as a black-box evaluator that computes centralized scoring signals over candidate decisions, and propose a Tacit Retrieval Memory (TRM) mechanism for implicit inter-agent coordination during decentralized execution.

**Compliance With Llm Reviewing Policy:**

Affirmed.

**Final Justification:**

I have confirmed my final score based on the first reviews and authors rebuttal.

**Key Questions For Authors:**

* Section 2: Preliminary and Motivation
1.	Action execution order among agents. The formulation introduces u_t as the set of actions taken by each agent at time step t. However, it is unclear how the execution ordering of these actions is determined. Do all agents execute their actions simultaneously at each time step? If so, this should be stated explicitly. If there is a sequential ordering, the mechanism for determining this order and its impact on coordination must be discussed.

* Section 3: Methodology
2.	Computing success/failure token probabilities before trajectory completion. The proposed method uses the LLM’s conditional probability of emitting success/failure tokens given the current trajectory and a candidate action as a centralized scoring signal. However, it is unclear how meaningful success/failure probabilities can be obtained at intermediate time steps when the trajectory has not yet terminated. The LLM has no access to ground-truth task outcomes at these points, so the resulting scores may reflect surface-level linguistic patterns rather than actual task progress. The authors should provide empirical evidence or theoretical justification for why these intermediate probability estimates serve as reliable value signals.
3.	Model requirements for Q_LLM computation. The joint LLM value Q_LLM is computed by a separate LLM that evaluates candidate joint actions. A critical question is whether this evaluator LLM must be a more capable (larger) model than the individual agent LLMs, or whether the same model can serve both roles. If the evaluator shares the same capacity as the agents, it is unclear why its centralized assessment would provide meaningful additional information beyond what each agent can already infer. The relationship between evaluator model capacity and the quality of the resulting coordination signals should be discussed.
4.	Scalability and coverage of the Tacit Retrieval Memory (TRM). This is a major concern. Several sub-questions arise regarding the TRM design:
(a)	Memory size: What is the storage size of b_i? How does it scale with the number of agents, the length of task trajectories, and the size of the action space?
(b)	Combinatorial coverage: Even for a single task, there can be numerous branching trajectories. During the centralized training (CT) phase, it is infeasible to enumerate all possible combinations of actions across all agents at every time step. How does the method handle this combinatorial explosion?
(c)	Missing entries at DE time: If the TRM does not contain a trajectory τ_{i,t} that matches the current decentralized execution context, what is the fallback mechanism? If retrieval is based on a similarity metric, what happens when no entry exceeds the similarity threshold? The paper should discuss this failure mode and its impact on coordination quality.
5.	Task decomposition into per-agent prompts. The final task objective is described as a single natural language sentence. However, the paper does not explain how this unified objective is decomposed into agent-specific sub-goals or prompts. Is this decomposition performed manually, by the LLM itself, or through some predefined template? This is a non-trivial step that significantly affects agent coordination quality and should be described in detail.
6.	Whether TRM access constitutes true decentralization. The paper claims that agents do not communicate with each other, achieving decentralized execution. However, every agent accesses the shared TRM during execution. The TRM is not stored locally within each agent’s LLM—it is a centralized resource that all agents query. This raises a fundamental question: can this architecture genuinely be characterized as decentralized execution? In the strict CTDE sense, decentralized execution requires that each agent’s policy depends solely on its own local observation history. Querying a shared external memory that was populated using joint trajectory data effectively introduces an indirect communication channel between agents. The authors should acknowledge this deviation from pure DE and discuss its implications, or provide a convincing argument for why shared static read-only memory does not violate the decentralization assumption.

* Section 4: TRM
1.	Insufficient detail on TRM construction (c and b generation). The mechanism for constructing and populating the TRM appears to be a central implementation component of the proposed method. However, the paper provides insufficient detail on how c and b are generated. What information is encoded in each? What model or procedure produces them? Without this level of detail, the contribution is difficult to assess and reproduce.

* Presentation Quality
1.	Redundancy across sections. The same motivations, contributions, and high-level ideas are repeated in the introduction, preliminary, methodology, and experiment sections. This significantly reduces the paper’s readability and wastes valuable space that could be used to address the technical gaps identified above. The authors should consolidate contributions in the introduction and devote the methodology section to precise technical exposition.

**Limitations:**

yes

**Strengths And Weaknesses:**

The paper addresses a practically relevant and theoretically interesting problem: enabling cooperation among closed-source LLM agents without explicit training or direct inter-agent communication. However, the paper suffers from significant redundancy in its presentation—the same contributions and motivations are repeated across nearly every section—and several core technical components lack sufficient clarity and justification. The following sections detail specific concerns organized by the paper’s structure.

---

> ### Author Rebuttal · Authors · 2026-03-31
>
> We thank the reviewer for the detailed and constructive feedback. Several concerns are already addressed in the manuscript; we clarify them below and provide new evidence where needed.
>
> **Q.1 Action execution order.**
>
> Our formulation assumes **synchronous joint actions**: at each step $t$, all agents independently output one action in parallel, and the environment advances using the combined $u_t=(u_t^1,\dots,u_t^N)$. No intra-step sequential ordering is assumed. This matches the **standard CTDE-style view** and is reflected in our decentralized VIKI-L2 setup (**Appendix D.2.6**), where independent per-robot planners run in parallel and outputs are merged by timestep alignment. We will add an explicit clarifying statement in the revision.
>
> **Q.2 Reliability of LLM critic.**
>
> We added a **critic reliability analysis** (**Table a4 in response to Reviewer zJkG**). Key findings: (1) the LLM critic ranks the expert action above the local alternative in **87.5%** of intermediate states vs. 48.7% for a random critic; (2) downstream task accuracy tracks critic quality monotonically; (3) adding Gaussian perturbations causes smooth degradation. These results confirm the intermediate critic signal carries substantial task-relevant information rather than reflecting surface-level linguistic patterns.
>
> **Q.3 Model requirements for the critic.**
>
> We do **not** require the critic to be larger than the agent LLMs. The critic's advantage comes from **centralized access** to the joint view, not from model scale. This parallels CTDE methods like QMIX [1]: the centralized component evaluates joint state/action information unavailable to decentralized agents. The same model family can serve both roles. Implementation details are in Appendix D.2.3.
>
> [1] QMIX: Monotonic Value Function Factorisation for Deep MARL
>
> **Q.4 TRM scalability and coverage.**
>
> (a) *Memory size*. $b_i(\tau_{i,t},\cdot)$ is per-agent, per-timestep over local candidates, so storage per timestep is $O(\sum_i|\mathcal{A}_{i,t}|)$. Only a filtered subset of informative steps is retained as TRM records. This cost is negligible in practice. Empirical statistics:
>
> | Benchmark | Total Records | Avg. per Agent | Size per Record |
> |-----------|-----:|-----:|-----:|
> | TDW-MAT | 1346 | 673 | 4.9 KB |
> | C-WAH | 1096 | 548 | 4.6 KB |
> | VIKI | 7763 | 3787 | 5.1 KB |
>
> (b) *Combinatorial explosion.* **Appendix D.2.4** describes our solution: we construct a small candidate set $\mathcal{U}(\tau_t)$ via one-sided counterfactuals rather than the full Cartesian product. Scoring cost scales as $\sum_i|\mathcal{A}_{i,t}|$ (linear in agents).
>
> (c) *Missing entries at test time.* If retrieval returns no sufficiently relevant entry, TRM context is simply not injected and the agent falls back to the base prompt. This failure mode is formalized as event $E_{\text{gap}}$ in **Proposition 4.1** and empirically quantified in **Table 7**—it is the dominant OOD failure source, confirming that TRM coverage directly determines coordination benefit.
>
> **Q.5 Task decomposition into per-agent prompts.**
>
> Our method does **not** perform explicit task decomposition. All agents receive the **same** shared task description (e.g., "Find and put 3 forks into the dishwasher"). Each agent's prompt is personalized only by agent identity and first-person local observation/history. Coordination is induced entirely through TRM: retrieved rules provide agent-specific guidance (e.g., one agent handles target objects, the other prioritizes complementary regions) without manually decomposing the task. VIKI follows the same principle **(Appendix D.2.6)**.
>
> **Q.6 Whether TRM violates decentralization.**
>
> TRM entries are constructed **per agent and per local progress**. At execution, agent $i$ queries TRM using only its **own local progress** $\tau_{i,t}$—the policy depends on local history plus a fixed contextual memory indexed by local history, not on teammates' current state. **Different agents retrieve complementary but separate contexts/rules** (**Figure 1/Figure 3: ingredient-chain vs. tool-chain**).
>
> TRM is **static and read-only** during execution, so it can be equivalently replicated locally to each agent before deployment, requiring no runtime centralized service. This is analogous to how CTDE methods distribute trained decentralized policies: the centralized phase produces agent-specific artifacts that are deployed locally.
>
> **Q.7 TRM details.**
>
> These are already specified: **$\mathbf{c}$** is the embedding of the retrieval key text, produced by mean-pooling last-layer hidden states of a text encoder (**Appendix D.2.5**). **$\mathbf{b}$** is the sparse fitted bias dictionary from the residual-fitting procedure (**Section 3.1 and Algorithm 1**). The retrieval pipeline is detailed in **Appendix D.2.5**.
>
> **Q.8 Presentation.**
>
> We appreciate this feedback. In the revision, we will consolidate high-level discussion and move more technical content from the appendix into the main methodology section.

---

> > ### Author Rebuttal · Reviewer_i9im · 2026-04-01
> >
> > My concerns have been adequately addressed.

---

> > > ### Author Response · Authors · 2026-04-03
> > >
> > > We thank the reviewer for the thoughtful feedback, for acknowledging the value of our work, and for raising the score. We also greatly appreciate the time and effort invested in reviewing our paper.

---

### Official Review · Reviewer_xbME · 2026-03-13

**Soundness:** 3
**Presentation:** 3
**Significance:** 2
**Originality:** 3
**Overall Recommendation:** 4
**Confidence:** 3

**Summary:**

This paper presents an approach to equip decentralized agents with tacit communication. The goal of tacit communication is to simulate the Centralized Training Decentralized Execution setup, wherein each agent executes actions based on a shared understanding of team value, thereby obviating the need for explicit communication between agents. The authors hypothesize that there must exist a nonzero subset of actions which maximize the global value of executing these actions as well as the product of local estimates by each decentralized agent. As such, the authors design an approach to minimize the bellman residual between the estimate of accumulated value with the “centralized” estimate of value (via an LLM). This delta is minimized and the resulting correction term tracking the compatibility of the greedy values with the global utility are stored in a memory bank as rules to dictate communication. These rules can then be retrieved through embedding-based similarity to allow an agent to plan using “tacit” context. The authors compare their approach to relevant baselines including fine-tuned models, as well as centralized and decentralized planners, on three benchmarks, to highlight the performance of Tact.

**Compliance With Llm Reviewing Policy:**

Affirmed.

**Key Questions For Authors:**

Included in strengths and weaknesses section.

**Limitations:**

Limitations section is missing.

**Strengths And Weaknesses:**

### Strengths

- I appreciated that the authors included a cost-analysis to empirically validate their claims regarding the high-cost flaw which ails the existing suite of communication methods which require message passing. Their cost analysis which shows that Tact generally requires less prompt/output tokens and model-calls, which is a strong motivating factor to adopt Tact in future work.
- The superior performance of Tact compared to a centralized baseline (across domains, and base-models) is a particularly strong result.
- The authors conduct thorough ablation studies to isolate the role of each component of their approach, demonstrating that combining these components in Tact leads to cumulative gains.

### Weaknesses

- Based on Eq-8, it seems as though, the approach requires knowledge of the optimal plan, u*. Is this correct? If so, is this optimal policy computed with a planner? There are many domains where an optimal planner does not exist, or cannot be assumed to exist. I’m curious to hear from the authors about how they would adapt their approach to such settings where an optimal plan is not available?
    - If my understanding is not correct, please feel free to correct me in the rebuttal.
- Minor - Mathematical notation is unclear/obtuse in some parts of the paper. I’ve listed a few examples below
    - In Eq 1, applying a product (\Pi) operation to an argmax term does not seem appropriate. Is this supposed to be a concatenation operation to accumulate the best greedy action across all agents?
    - In Eq 2, I wasn’t sure what the winning and losing actions, y_w and y_l, were. These terms were not discussed previously or at any point after this equation.

---

> ### Author Rebuttal · Authors · 2026-03-31
>
> We thank the reviewer for the constructive review. We are encouraged that you found our cost analysis valuable, recognized the strong performance of Tact over the baselines, and appreciated the thorough ablations showing the cumulative contribution of each component. Below, we respond to the weaknesses you raised.
>
> **W.1**
> Our method does not require access to an oracle optimal planner in general environments. **Appendix D.2.4** already clarifies this point. In C-WAH and TDW-MAT, the optimal joint action is not available. In these settings, our method does not rely on an oracle planner. Instead, we use the actually executed joint action as the anchor, and construct a one-sided counterfactual candidate set for centralized scoring.
>
> The theoretical guarantee is stated relative to the finite candidate set $\mathcal{U}(\tau)$ under the joint evaluator, i.e., with respect to the best item within the scored candidate set. It does not assume access to a globally optimal policy over the full environment dynamics. In this sense, the method only requires a reasonable anchor joint action and a tractable candidate set for offline centralized evaluation, rather than an oracle optimal planner for the whole domain.
>
> **W.2**
> $\prod$ denotes the Cartesian product, which is standard notation in the Dec-POMDP literature: each agent independently selects its optimal action, and $\prod$ represents the Cartesian product of these per-agent argmax sets, which is then intersected with the joint argmax set. This usage is standard in [1][2].
> Our intention was not numeric multiplication, to avoid confusion, we will revise it to a more explicit Cartesian-product notation.
>
>
> $y_w := \texttt{success}$ and $y_l := \texttt{fail}$ denote the designated success and failure outcome tokens, respectively. We note that the paper already states after Eq.(2) that $y_w$ and $y_l$ are success-failure tokens. We will make this definition more explicit in the revised version.
>
> [1] Dual Self-Awareness Value Decomposition Framework without Individual Global Max for Cooperative MARL
> [2] DFAC Framework: Factorizing the Value Function via Quantile Mixture for Multi-Agent Distributional Q-Learning

---

> > ### Author Rebuttal · Reviewer_xbME · 2026-04-08
> >
> > Thank you for your rebuttal. Your response addressed some of my concerns, however, I still retain my original assessment of the paper.

---

### Official Review · Reviewer_GV8T · 2026-03-25

**Soundness:** 1
**Presentation:** 2
**Significance:** 2
**Originality:** 3
**Overall Recommendation:** 4
**Confidence:** 3

**Summary:**

This work studies multi-agent coordination in robot systems that leverage LLMs as policies. As opposed to previous works that use a centralized planner or linguistic communication, this work proposes to learn a shared context to enable multi-agent coordination. The proposed method, TACT, first learns a global or joint value function $Q_{LLM}$ over all agents by using an LLM-as-a-judge to guess whether an action will lead to successful trajectories (whether the LLM outputs "success" or "fail"). They then learn a local per-agent value function $q_{LLM}$ in the same way. They propose to learn a "bias" that accounts for the difference between global and local values ("residual") using an EM-like iterative algorithm they term *residual banding and bias fitting*. It is a simple method that sets the same bias $b$ for all agents to make sure that the sum of local agent values + $b$ is within some band $\epsilon$ of the global joint value.

The final step is taking the bias $b$ and converting it into a textual prompt, used to condition the policy at test time. This is done by finding the states in trajectories with highest/lowest corresponding bias and prompting an LLM to create a rule to priotize/avoid these actions based on the positive/negative bias. These rules are stored in a *TRM*. At test time, agents embed their current trajectory to retrieve the most similar rule to their current situation and use it in their prompt.

They test on one robot RL benchmark, VIKI-L2, and two zero-shot robot benchmarks C-WAH and TDW. TACT outperforms basic non-training baselines on VIKI-L2. It outperforms zero-shot coordination baselines in C-WAH and TDW. TACT is also more token-efficient than zero-shot coordination through communication with CoELA and an ablation shows that each component is necessary for full performance. The method is reasonably robust to choice of band $\epsilon$.

**Compliance With Llm Reviewing Policy:**

Affirmed.

**Final Justification:**

### author responses
The authors have promised to remove "training-free" from their description, opting for something closer to the actual "gradient-free prompt optimization" nature of their method.

They have argued their baselines do a fair amount of manual prompt optimization

They point out that their gradient-free optimization makes it suitable for closed-source black-box models.

They have also compared to a decentralized RL baseline, and although I have issues with this specific experiment, I am reasonably convinced their overall method has merit.

They corrected my misunderstanding of bias-fitting.

### summary

Overall I feel that the method is still made up of many parts that are not sufficiently well ablated (e.g. the training paradigm vs the LLM-as-a-judge feedback mechanism) but I believe that the combined method has merit, is applicable to closed-source models, and can achieve non-zero reward in a difficult VIKI-L2 scenario.

**Key Questions For Authors:**

You refer to your "per-agent bias $b_i$ but since you evenly distribute errors, aren't all "per-agent" biases the same, so shouldn't it be just a "local bias $b$" ? If all per-agent biases are the same, then how is it possible that your TRM instructions differ for the agents e.g. role? Shouldn't those rules be conditioned on purely trajectory history then?

Your method performs worse than the VIKI-R RL baseline on VIKI despite it using a much smaller 3B model. Can you compare to RL training or explain when would you be able to use your method and not RL?

Why do positive/negative biases correspond to encourage/avoid? Shouldn't biases correspond to *relative* over/underestimations of the value and therefore correspond to uncertainty? I would assume that encourage/avoid should be based on *absolute* values.

By "tacit communication" do you mean "non-verbal/non-linguistic coordination"? I think "coordinating through shared context" is a much clearer way to express your idea.

Is there a justification of your assumption of evenly distributed errors across actions? (Equation 9)

**Limitations:**

yes

**Strengths And Weaknesses:**

### weaknesses
Overall I think soundness is greatly missing from this work, which in turn affects its significance.

The main issue is that the method is compared to zero-shot baselines and advertises itself as a "training-free" protocol, but the proposal is literally a training method (a search method for a protocol though it isn't gradient-based) that explicitly requires trajectories in the environment from these policies. Since it is a training method, I feel it should be compared to other training methods i.e. RL and at least on VIKI bench, it is worse.

The other issue is that the method is seemingly a bag of tricks without great justification and the intermediate steps are never verified e.g. are the learned values accurate? do the local biases correspond to something useful? how well does a completely random LLM generating rules do compared with your bias-conditioned LLM? There are better results on these benchmarks in the literature so a bag of tricks isn't de-facto impactful since it isn't SOTA.

The C-WAH and TDW-MAT baselines are fair comparisons because none of them (ProAgent, CoELA, RoCo) do any training. They are zero-shot works whereas TACT explicitly trains on trajectories.

The paper is a little confusing written. The authors propose a lot of new things and notation can be unclear e.g. they describe creating TRM entries $e$ and never use the notation $e$ again. The entries $e$ store a bias dictionary $b$ in them but it isn't explained how this is used. The other two parts of $e$, $r$ and $c$ are clear, but also that shorthand is never used again.

### strengths
The originality of an LLM-based local vs global value function is quite intriguing and could be impactful. Sadly, the value functions are never verified themselves for accuracy.

The benchmarks are well-chosen and results seem to reproduce other previous work. The problem of multi-robot coordination is an interesting one. Figure 1 has a lot going on but it is actually quite informative. The appendix and algorithm section did a lot to clarify the overall method as well. The ablations are very convincing about the need for all parts of the algorithm and do a lot to inspire confidence that this method works.

---

> ### Author Rebuttal · Authors · 2026-03-31
>
> We appreciate the reviewer for the thoughtful review. Some concerns appear to arise from misunderstandings of our method. We address each concern below.
>
> **W.1 "Training-free" claim.**
>
> (a) By *training-free* we mean **no model weights are updated**, which is the standard meaning in the LLM-agent literature. Our offline rule-mining stage is analogous to RAG database construction, not model training.
>
> Importantly, the baselines are also **not strictly zero-shot**:
>
> | Method | Gradients | Task-Specific Eng. | Env. Interaction |
> |---|---|---|---|
> | ProAgent | ✗ | ✓ (Knowledge Library, Verificator) | ✓ (multi-round verification) |
> | RoCo | ✗ | ✓ (per-robot dialog templates) | ✓ (iterative dialog + collision feedback) |
> | CoELA | ✗ | ✓ (communication module) | ✓ (online messaging overhead) |
> | **Tact** | ✗ | ✗ (rules auto-generated) | ✓ (offline bias fitting) |
> | RL | ✓ | ✗ | ✓ (extensive rollouts) |
>
> Tact replaces hand-crafted prompt engineering with an automated, data-driven process.
>
> (b) **VIKI-R is a centralized RL policy** with global-state access, whereas our setting is **decentralized and partially observable**. The relevant question is whether Tact **helps decentralized executors coordinate**, not whether it outperforms a global controller. We added an experiment using RL as the decentralized executor with Tact rules:
>
> | Qwen2.5-VL-7B-RL | P-cen (RL) | P-dec | **Tact** |
> |---|---:|---:|---:|
> | ACC_ID ↑ | 95.2 | 9.4 | **96.8** |
> | ACC_OOD ↑ | 33.2 | 2.8 | **35.3** |
>
> Tact recovers near-centralized performance from a decentralized executor. Tact and RL are **complementary**: Tact applies to closed-source LLMs where weight updates are infeasible; RL excels when fine-tuning is possible.
>
> **W.2 Intermediate verification.**
>
> Our **ablation (Table 4)** systematically validates each component—removing any one causes catastrophic failure, which is inconsistent with a "bag of tricks."
>
> (1) *Are local biases useful? **Appendix C.3, Table 7** reports a direction-recovery audit: full TRM achieves a mismatch rate of 0.28 vs. 0.75 without bias supervision, confirming the fitted bias encodes meaningful coordination signals.
>
> (2) *How does random rule generation compare? **Table 4 row A3** keeps retrieval but removes bias supervision, making rule generation effectively free-form. A3 collapses to near-zero OOD accuracy, showing that ungrounded rules are ineffective.
>
> (3) *Are the learned values accurate? We added a critic reliability analysis (**Table a4 in our response to Reviewer zJkG**): the LLM critic ranks the expert action above the local alternative in 87.5% of states, and downstream accuracy tracks critic quality monotonically.
>
> **W.3 Baseline fairness**
>
> Please refer to W.1 above.
>
> **W.4 Notation clarity.**
>
> The entry shorthand $e$ and its fields $(r,\mathbf{c},\mathbf{b})$ are reused in **Appendix A.2**, where we explicitly reference $e.\mathbf{b}$ to define action coverage and directional agreement. **Appendix D.2.5** further details TRM implementation (retrieval key, encoder, similarity). At execution time the agent consumes the retrieved rule text $\hat{x}_{i,t}$, so the exposition naturally shifts from entry-level notation to its consumed output.
>
> **Q.1 Per-agent biases are not identical.**
>
> The equal split in Eq. 9 applies to one specific joint action $u$, **not** to the final bias tables. Bias fitting sweeps over the entire candidate set $\mathcal{U}(\tau_t)$ for multiple rounds (**Algorithm 1**). For joint action $u=(a_1,\dots,a_N)$, agent $i$ updates only $b_i(\tau_i,a_i)$. Since different joint actions assign different local actions to different agents, **the accumulated biases diverge across agents**. This agent-specific asymmetry produces differentiated TRM rules.
>
> **Q.2 Comparison with RL.**
>
> Please refer to W.1.
>
> **Q.3 Why positive/negative bias → encourage/avoid.**
>
> The bias is a **correction to align local values with joint values**, **not an uncertainty estimate**. A positive bias means the local evaluator underestimates action $a$'s contribution to team success—the joint evaluator scores plans containing $a$ higher than the agent infers locally—so $a$ should be encouraged. The sign encodes a directional coordination correction.
>
> **Q.4 "tacit communication."**
>
> We use it in the game-theoretic sense: coordination through shared conventions and common knowledge rather than explicit message exchange. This connects to an established tradition, and **Definition 2.1** formalizes its specific meaning in our setting.
>
> **Q.5 Even error distribution.**
>
> Eq. 9 is a symmetric credit-assignment step for a single joint candidate, **not** a claim that credit is identical across agents. In practice, this local split is progressively refined over the full sweep on $\mathcal{U}(\tau_t)$, where different joint actions update different entries for different agents. The approach is in spirit similar to early value-decomposition heuristics [1].
>
> [1] Value-Decomposition Networks For Cooperative Multi-Agent Learning

---

> > ### Author Rebuttal · Reviewer_GV8T · 2026-04-01
> >
> > Thank you for your response.
> >
> > My major concerns about comparison to RL has not been resolved but my other concerns are.
> >
> > **TACT is not Training-Free**
> >
> > Your method uses data from execution traces to optimize the prompt.
> > This is de-facto training and essentially prompt optimization.
> > Weights do not have to change for it to be "training".
> >
> > "Black box optimization" is perhaps the term you're looking for.
> >
> > **Your comparison to baselines is incorrect**
> >
> > None of the baselines use data from execution traces. Your method gets extra data by generating trajectories on the benchmark.
> >
> > Your baselines *do* some task-specific engineering but *so does TACT* with task-specific prompt templates.
> >
> > Overall, you method does training, whereas your baselines do not. This isn't bad on its own but you need to compare to a baselines that train.
> >
> > **Comparison to RL**
> >
> > I do not necessarily expect your method to outperform centralized training (VIKI-R) or even to outperform decentralized RL. I do expect comparing to *some* sort of baseline that trains on data because otherwise your method simply uses execution data that is not given to your baselines.
> >
> > Can you, at minimum, report how many trajectories you use for bias fitting and how this compares to RL? Sample efficiency is another way your method can distinguish itself.
> >
> > The ideal experiment is to run decentralized RL on the same number of trajectories you use for bias-fitting to demonstrate how your black-box method compares to white-box RL.
> >
> > **Clarification**
> >
> > Just to confirm, your experiments above are not doing any RL training? You are using the RL-trained model from VIKI?
> >
> > **Resolved concern: per-agent biases**
> >
> > Thank you for the clarification, this makes sense. I might suggest changing Algorithm 1 to change
> > $b_i(\tau_i,t, a) \gets 0$
> > into
> > $b_i(\tau_i,t, a) \gets {0,0,...0} $ to make it clear that it is a vector with biases per action, not a scalar

---

> > > ### Author Response · Authors · 2026-04-03
> > >
> > > We sincerely thank the reviewer for the continued engagement and constructive feedback. We address each point below.
> > >
> > > **1."Training-free" terminology**
> > >
> > >
> > >
> > > Our motivation was to emphasize that TACT does not perform gradient-based optimization, and that its offline warm-up stage has modest computational overhead, as further supported by the additional experiments in this response below.
> > >
> > > Nevertheless, TACT does require an offline trajectory-dependent stage for bias fitting and rule distillation. We will revise the terms to make this motivation explicit and avoid the confusion.
> > >
> > >
> > >
> > >
> > > **2.Baseline comparison**
> > >
> > >
> > >
> > > We would like to clarify that, although these baselines do not use trajectory data, they are not cost-free: each of them relies on substantial task-specific engineering, and it is difficult to make all methods perfectly aligned in engineering effort. For example, ProAgent requires hand-crafted Knowledge Libraries, task-specific state-grounding code, and custom Verificator prompts for each domain; RoCo relies on a centralized motion planner with iterative collision feedback; and CoELA requires a dedicated communication module. In contrast, TACT extracts coordination knowledge through an algorithmically automated, data-driven process.
> > >
> > > At the same time, we agree with the reviewer that comparing against a method that also learns from trajectory data would strengthen the evaluation. Following this suggestion, we added a training-based decentralized RL baseline, since most existing embodied multi-agent LLM methods are still prompt-engineering based and do not provide such a comparison. The new results show that TACT achieves better sample efficiency than the decentralized RL baseline, while also obtaining stronger performance. Therefore, TACT can use trajectory data in a much lighter-weight way to extract coordination knowledge, without incurring the full cost of decentralized RL training.
> > >
> > >
> > >
> > > **3.Comparison to RL**
> > >
> > > We thank the reviewer for this suggestion. We trained a **decentralized RL baseline** following the decentralized setup, where each agent's policy predicts actions based solely on its own local observation. We call this baseline **Dec-RL-ZERO**, trained on Qwen2.5-VL-3B using GRPO for 15 epochs with batch size 256 and rollout $n$=5, covering 3,765 task trajectories (~8 GPU-hours).
> > >
> > > **Table a5. Sample efficiency comparison on VIKI-L2.**
> > >
> > > | Method | Fitting Traj. | Total Env Steps | Sampled LLM Outputs | ACC_ID | ACC_OOD |
> > > |---|---:|---:|---:|---:|---:|
> > > | **TACT (3B)** | 3,765 | 26,004 | 158600 | 8.45 | 0.00 |
> > > | **Dec-RL-ZERO (3B)** | 3,765 | 26,004 | 578,550| 0.72 | 0.00 |
> > > | **VIKI-R-ZERO (3B)** | 7,196 | 48,523 | ~539,700 | 0.00 | 0.00 |
> > > | **VIKI-R-ZERO (7B)** | 7,196 | 48,523 | ~539,700 | 0.17 | 0.00 |
> > >
> > > TACT and Dec-RL-ZERO use the same base model without SFT to ensure a controlled comparison under the same model capacity and data budget. VIKI-R-ZERO is the centralized RL method without the SFT stage, we additionally include the official VIKI-R-ZERO 3B/7B results as centralized RL references.
> > >
> > > To make the accounting explicit: for RL, Sampled LLM Outputs = trajectories × epochs × GRPO rollout $n$ × agents; for TACT, we count the total number of offline value queries, i.e., $\sum_t\left|\mathcal{U}\left(\tau_t\right)\right|$ joint-action scoring queries plus per-agent local value queries. TACT consumes roughly **one quarter** of the sampled outputs compared to RL baselines, yet achieves better in-distribution accuracy than Dec-RL-ZERO. This demonstrates that TACT offers better sample efficiency than decentralized RL under the same trajectory budget.
> > >
> > >
> > >
> > > **4.Clarification**
> > >
> > > Yes, in our W.1 response, we used the officially released VIKI-R checkpoint, no new RL training was performed there. In the new comparison above (Table a5), we additionally trained our own decentralized RL baseline (Dec-RL-ZERO) from scratch to provide the controlled comparison the reviewer suggested.
> > >
> > >
> > >
> > > **5.Notation in Algorithm 1**
> > >
> > > We will update Algorithm 1 to clarify that in the revision. We are glad that our earlier clarification resolved the reviewer‘s concerns, and we sincerely appreciate the reviewer's constructive engagement throughout this discussion.

---

### Decision · Program_Chairs · 2026-04-30

**Decision:**

Accept (regular)

**Comment:**

reviewers agree Tact addresses a critical challenge in decentralized LLM-agent cooperation through a novel contextual tacit communication framework that bypasses explicit message passing [xbME, zJkG]. The system achieves strong empirical results across three benchmarks (i.e., VIKI, C-WAH, TDW-MAT) and outperforms zero-shot baselines while significantly reducing token and inference costs compared to communication-heavy methods.

Reviewers raised concerns regarding the "training-free" terminology, the lack of RL baselines, the reliability of the intermediate LLM critic, and the potential requirement of an oracle planner [GV8T, i9im, xbME]. In the rebutal, the authors provided a newly trained decentralized RL baseline, added a critic reliability analysis, and clarified the theoretical assumptions. reviewers GV8T and zJkG noted the rebuttal resolved their primary concerns and raised their scores to weak accept to match the consensus.